# Accuracy of novel antigen rapid diagnostics for SARS-CoV-2: A living systematic review and meta-analysis

Lukas E. Brümmer[1☯], Stephan Katzenschlager[2☯], Mary Gaeddert[1], Christian Erdmann[3], Stephani Schmitz[1], Marc Bota[4], Maurizio Grilli[5], Jan Larmann[2], Markus A. Weigand[2], Nira R. Pollock[6], Aurélien Macé[7], Sergio Carmona[7], Stefano Ongarello[7], Jilian A. Sacks[7], Claudia M. Denkinger[1,8]*

1 Division of Tropical Medicine, Center for Infectious Diseases, Heidelberg University Hospital, Heidelberg, Germany, 2 Department of Anesthesiology, Heidelberg University Hospital, Heidelberg, Germany, 3 FH Muenster University of Applied Sciences, Muenster, Germany, 4 Agaplesion Bethesda Hospital, Hamburg, Germany, 5 Library, University Medical Center Mannheim, Mannheim, Germany, 6 Department of Laboratory Medicine, Boston Children's Hospital, Boston, Massachusetts, United States of America, 7 FIND, Geneva, Switzerland, 8 Partner Site Heidelberg University Hospital, German Center for Infection Research (DZIF), Heidelberg, Germany

☯ These authors contributed equally to this work.
* Claudia.Denkinger@uni-heidelberg.de

**Data Availability Statement:** All raw data is publicly available under https://zenodo.org/record/4924035#.YMSDOS2230o.

## Abstract

### Background

SARS-CoV-2 antigen rapid diagnostic tests (Ag-RDTs) are increasingly being integrated in testing strategies around the world. Studies of the Ag-RDTs have shown variable performance. In this systematic review and meta-analysis, we assessed the clinical accuracy (sensitivity and specificity) of commercially available Ag-RDTs.

### Methods and findings

We registered the review on PROSPERO (registration number: CRD42020225140). We systematically searched multiple databases (PubMed, Web of Science Core Collection, medRvix, bioRvix, and FIND) for publications evaluating the accuracy of Ag-RDTs for SARS-CoV-2 up until 30 April 2021. Descriptive analyses of all studies were performed, and when more than 4 studies were available, a random-effects meta-analysis was used to estimate pooled sensitivity and specificity in comparison to reverse transcription polymerase chain reaction (RT-PCR) testing. We assessed heterogeneity by subgroup analyses, and rated study quality and risk of bias using the QUADAS-2 assessment tool. From a total of 14,254 articles, we included 133 analytical and clinical studies resulting in 214 clinical accuracy datasets with 112,323 samples. Across all meta-analyzed samples, the pooled Ag-RDT sensitivity and specificity were 71.2% (95% CI 68.2% to 74.0%) and 98.9% (95% CI 98.6% to 99.1%), respectively. Sensitivity increased to 76.3% (95% CI 73.1% to 79.2%) if analysis was restricted to studies that followed the Ag-RDT manufacturers' instructions. LumiraDx showed the highest sensitivity, with 88.2% (95% CI 59.0% to 97.5%). Of instrument-free Ag-RDTs, Standard Q nasal performed best, with 80.2% sensitivity (95% CI

**Funding:** The study was supported by the Ministry of Science, Research and Arts of the State of Baden-Wuerttemberg, Germany (no grant number; https://mwk.baden-wuerttemberg.de/de/startseite/) and internal funds from the Heidelberg University Hospital (no grant number; https://www.heidelberg-university-hospital.com/de/) to CMD. Further, this project was funded by United Kingdom (UK) aid from the British people (grant number: 300341-102; Foreign, Commonwealth & Development Office (FCMO), former UK Department of International Development (DFID); www.gov.uk/fcdo), and supported by a grant from the World Health Organization (WHO; no grant number; https://www.who.int) and a grant from Unitaid (grant number: 2019-32-FIND MDR; https://unitaid.org) to Foundation of New Diagnostics (FIND; JAS, SC, SO, AM). The funders had no role in study design, data collection and analysis, decision to publish, or preparation of the manuscript.

**Competing interests:** I have read the journal's policy and the authors of this manuscript have the following competing interests: CMD is a member of the Editorial Board of PLOS Medicine.

**Abbreviations:** Ag-RDT, antigen rapid diagnostic test; AN, anterior nasal; BAL/TW, bronchoalveolar lavage and throat wash; CI, confidence interval; Ct, cycle threshold; IFU, instructions for use; LOD, limit of detection; MT, mid-turbinate; NP, nasopharyngeal; OP, oropharyngeal; PFU, plaque forming units; RT-PCR, reverse transcription polymerase chain reaction.

70.3% to 87.4%). Across all Ag-RDTs, sensitivity was markedly better on samples with lower RT-PCR cycle threshold (Ct) values, i.e., <20 (96.5%, 95% CI 92.6% to 98.4%) and <25 (95.8%, 95% CI 92.3% to 97.8%), in comparison to those with Ct $\geq$ 25 (50.7%, 95% CI 35.6% to 65.8%) and $\geq$30 (20.9%, 95% CI 12.5% to 32.8%). Testing in the first week from symptom onset resulted in substantially higher sensitivity (83.8%, 95% CI 76.3% to 89.2%) compared to testing after 1 week (61.5%, 95% CI 52.2% to 70.0%). The best Ag-RDT sensitivity was found with anterior nasal sampling (75.5%, 95% CI 70.4% to 79.9%), in comparison to other sample types (e.g., nasopharyngeal, 71.6%, 95% CI 68.1% to 74.9%), although CIs were overlapping. Concerns of bias were raised across all datasets, and financial support from the manufacturer was reported in 24.1% of datasets. Our analysis was limited by the included studies' heterogeneity in design and reporting.

## Conclusions

In this study we found that Ag-RDTs detect the vast majority of SARS-CoV-2-infected persons within the first week of symptom onset and those with high viral load. Thus, they can have high utility for diagnostic purposes in the early phase of disease, making them a valuable tool to fight the spread of SARS-CoV-2. Standardization in conduct and reporting of clinical accuracy studies would improve comparability and use of data.

## Author summary

### Why was this study done?

- Antigen rapid diagnostic tests (Ag-RDTs) are considered an important diagnostic tool to fight the spread of SARS-CoV-2.

- An increasing number of Ag-RDTs are offered on the market, and a constantly growing body of literature evaluating their performance is available.

- To inform decision makers about the best test to choose, an up-to-date summary of their performance is needed.

### What did the researchers do and find?

- On a weekly basis, we search multiple databases for evaluations of Ag-RDTs detecting SARS-CoV-2 and post the results on https://www.diagnosticsglobalhealth.org.

- Based on the search results up until 30 April 2021, we conducted a systematic review and meta-analysis, including a total of 133 clinical and analytical accuracy studies.

- Across all meta-analyzed studies, when Ag-RDTs were performed according to manufacturers' recommendations, they showed a sensitivity of 76.3% (95% CI 73.1% to 79.2%), with LumiraDx (sensitivity 88.2% [95% CI 59.0% to 97.5%]) and, of the instrument-free Ag-RDTs, Standard Q nasal (74.9% sensitivity [95% CI 69.3% to 79.7%]) performing best.

- Across all Ag-RDTs, sensitivity increased to 95.8% (95% CI 92.3% to 97.8%) when we restricted the analysis to samples with high viral loads (i.e., a Ct value < 25) and to 83.8% (95% CI 76.3% to 89.2%) when tests were performed on patients within the first week after symptom onset.

### What do these findings mean?

- Ag-RDTs detect the vast majority of cases within the first week of symptom onset and those with high viral load. Thus, they can have high utility for diagnostic purposes in the early phase of disease.

- Out of all assessed tests, LumiraDx showed the highest accuracy. Standard Q was the best-performing test when only considering those that do not require an instrument.

- A standardization of reporting methods for clinical accuracy studies would enhance future test comparisons.

## Introduction

As the COVID-19 pandemic continues around the globe, antigen rapid diagnostic tests (Ag-RDTs) for SARS-CoV-2 are seen as an important diagnostic tool to fight the virus's spread [1,2]. The number of Ag-RDTs on the market is increasing constantly [3]. Initial data from independent evaluations suggest that the performance of SARS-CoV-2 Ag-RDTs may be lower than what is reported by the manufacturers. In addition, Ag-RDT accuracy seems to vary substantially between tests [4–6].

With the increased availability of Ag-RDTs, an increasing number of independent validations have been published. Such evaluations differ widely in their quality, methods, and results, making it difficult to assess the true performance of the respective tests [7]. To inform decision makers on the best choice of individual tests, an aggregated, widely available, and frequently updated assessment of the quality, performance, and independence of the data is urgently needed. While other systematic reviews have been published, they include data only up until November 2020 [8–11], exclude preprints [12], or were industry sponsored [13]. In addition, only 1 assessed the quality of studies in detail, with data up until November 2020 [7,11].

With our systematic review and meta-analysis, we aim to close this gap in the literature and link to a website (https://www.diagnosticsglobalhealth.org) that is regularly updated.

## Methods

We developed a study protocol following standard guidelines for systematic reviews [14,15], which is available in S1 Text. We also completed the PRISMA checklist (S1 PRISMA Checklist). Furthermore, we registered the review on PROSPERO (registration number: CRD42020225140).

### Search strategy

We performed a search of the databases PubMed, Web of Science, medRxiv, and bioRxiv using search terms that were developed with an experienced medical librarian (M. Grilli) using combinations of subject headings (when applicable) and text-words for the concepts of the search question. The main search terms were "Severe Acute Respiratory Syndrome

Corona-virus 2," "COVID-19," "Betacoronavirus," "Coronavirus," and "Point of Care Test-
ing." The full list of search terms is available in S2 Text. We also searched the Foundation for
Innovative New Diagnostics (FIND) website (https://www.finddx.org/sarscov2-eval-antigen/)
for relevant studies manually. We performed the search up until 30 April 2021. No language
restrictions were applied.

### Inclusion criteria

We included studies evaluating the accuracy of commercially available Ag-RDTs to establish a
diagnosis of SARS-CoV-2 infection, against reverse transcription polymerase chain reaction
(RT-PCR) or cell culture as reference standard. We included all study populations irrespective
of age, presence of symptoms, or study location. We considered cohort studies, nested cohort
studies, case–control or cross-sectional studies, and randomized studies. We included both
peer-reviewed publications and preprints.

We excluded studies in which patients were tested for the purpose of monitoring or ending
quarantine. Also, publications with a population size smaller than 10 were excluded. Although
the size threshold of 10 is arbitrary, such small studies are more likely to give unreliable esti-
mates of sensitivity and specificity.

### Index tests

Ag-RDTs for SARS-CoV-2 aim to detect infection by recognizing viral proteins. Most Ag-
RDTs use specific labeled antibodies attached to a nitrocellulose matrix strip, to capture the
virus antigen. Successful binding of the antibodies to the antigen either is detected visually
(through the appearance of a line on the matrix strip [lateral flow assay]) or requires a specific
reader for fluorescence detection. Microfluidic enzyme-linked immunosorbent assays have
also been developed. Ag-RDTs typically provide results within 10 to 30 minutes [6].

### Reference standard

Viral culture detects viable virus that is relevant for transmission but is available in research
settings only. Since RT-PCR tests are more widely available and SARS-CoV-2 RNA (as
reflected by RT-PCR cycle threshold [Ct] value) highly correlates with SARS-CoV-2 antigen
quantities, we considered it an acceptable reference standard for the purposes of this system-
atic review [16]. It is of note that there is currently no international standard for the classifica-
tion of viral load available.

### Study selection and data extraction

Two reviewers (LEB and CE, LEB and SS, or LEB and MB) reviewed the titles and abstracts of
all publications identified by the search algorithm independently, followed by a full-text review
for those eligible, to select the articles for inclusion in the systematic review. Any disputes were
solved by discussion or by a third reviewer (CMD).

A full list of the parameters extracted is included in S1 Table, and the data extraction file is
available at https://zenodo.org/record/4924035#.YOlzWS223RZ. Studies that assessed multiple
Ag-RDTs or presented results based on differing parameters (e.g., various sample types) were
considered as individual datasets.

At first, 4 authors (SK, CE, SS, and MB) extracted 5 randomly selected papers in parallel to
align data extraction methods. Afterwards, data extraction and the assessment of methodologi-
cal quality and independence from test manufacturers (see below) was performed by 1 author

per paper (SK, CE, SS, or MB) and controlled by a second (LEB, SK, SS, or MB). Any differences were resolved by discussion or by consulting a third author (CMD).

## Study types

We differentiated between clinical accuracy studies (performed on clinical samples) and analytical accuracy studies (performed on spiked samples with a known quantity of virus). Analytical accuracy studies can differ widely in methodology, impeding an aggregation of their results. Thus, while we extracted the data for both kinds of studies, we only considered data from clinical accuracy studies as eligible for the meta-analysis. Separately, we summarized the results of analytical studies and compared them with the results of the meta-analysis for individual tests.

## Assessment of methodological quality

The quality of the clinical accuracy studies was assessed by applying the QUADAS-2 tool [17]. The tool evaluates 4 domains: patient selection, index test, reference standard, and flow and timing. For each domain, the risk of bias is analyzed using different signaling questions. Beyond the risk of bias, the tool also evaluates the applicability of each included study to the research question for every domain. The QUADAS-2 tool was adjusted to the needs of this review and can be found in S3 Text.

## Assessment of independence from manufacturers

We examined whether a study received financial support from a test manufacturer (including the free provision of Ag-RDTs), whether any study author was affiliated with a test manufacturer, and whether a respective conflict of interest was declared. Studies were judged not to be independent from the test manufacturer if at least 1 of these aspects was present; otherwise, they were considered to be independent.

## Statistical analysis and data synthesis

We extracted raw data from the studies and recalculated performance estimates where possible based on the extracted data. The raw data can be found in S2 Table. We prepared forest plots for the sensitivity and specificity of each test and visually evaluated the heterogeneity between studies. If 4 or more datasets were available with at least 20 positive RT-PCR samples per dataset for a predefined analysis, a meta-analysis was performed. We report point estimates of sensitivity and specificity for SARS-CoV-2 detection compared to the reference standard along with 95% confidence intervals (CIs) using a bivariate model (implemented with the "reitsma" command from the R package "mada," version 0.5.10). When there were fewer than 4 studies for an index test, only a descriptive analysis was performed, and accuracy ranges are reported. In subgroup analyses where papers presented data only on sensitivity, a univariate random-effects inverse variance meta-analysis was performed (using the "metagen" command from the R package "meta," version 4.11–0). We predefined subgroups for meta-analysis based on the following characteristics: Ct value range, sampling and testing procedure in accordance with manufacturer's instructions as detailed in the instructions for use (IFU) (henceforth called IFU-conforming) versus not IFU-conforming, age (<18 versus ≥18 years), sample type, presence or absence of symptoms, symptom duration (<7 days versus ≥7 days), viral load, and type of RT-PCR used.

In an effort to use as much of the heterogeneous data as possible, the cutoffs for the Ct value groups were relaxed by 2–3 points within each range. The <20 group included values reported

up to ≤20, the <25 group included values reported as ≤24 or <25 or 20–25, and the <30 group included values from ≤29 to ≤33 and 25–30. The ≥25 group included values reported as ≥25 or 25–30, and the ≥30 group included values from ≥30 to ≥35. For the same reason, when categorizing by age, the age group <18 years (children) included samples from persons whose age was reported as <16 or <18 years, whereas the age group ≥18 years (adults) included samples from persons whose age was reported as ≥16 years or ≥18.

For categorization by sample type, we assessed (1) nasopharyngeal (NP) alone or combined with other (e.g., oropharyngeal [OP]), (2) OP alone, (3) anterior nasal (AN) or mid-turbinate (MT), (4) a combination of bronchoalveolar lavage and throat wash (BAL/TW), or (5) saliva. Analyses were preformed using R 4.0.3 (R Foundation for Statistical Computing, Vienna, Austria).

We aimed to do meta-regression to examine the impact of covariates including symptom duration and Ct value range. We also performed the Deeks test for funnel-plot asymmetry as recommended to investigate publication bias for diagnostic test accuracy meta-analyses [18] (using the "midas" command in Stata, version 15); a $p$-value < 0.10 for the slope coefficient indicates significant asymmetry.

### Sensitivity analysis

Two types of sensitivity analyses were planned: estimation of sensitivity and specificity excluding case–control studies, and estimation of sensitivity and specificity excluding non-peer-reviewed studies. We compared the results of each sensitivity analysis against the overall results to assess the potential bias introduced by considering case–control studies and non-peer-reviewed studies.

## Results

### Summary of studies

The systematic search resulted in 14,254 articles. After removing duplicates, 8,921 articles were screened, and 266 papers were considered eligible for full-text review. Of these, 148 were excluded because they did not present primary data [13,19–131] or the Ag-RDT was not commercially available [16,132–164], leaving 133 studies to be included in the systematic review (Fig 1) [4,165–296].

At the end of the data extraction process, 37 studies were still in preprint form [4,171,173,174,177,180,190,192,201,204,205,207,211,214–216,218,220,222,223,225,227,231, 233,234,238,240,244,247,253,257,265,267,284,287,290,293]. All studies were written in English, except for 2 in Spanish [175,280]. Out of the 133 studies, 9 reported analytical accuracy [173,191,198,208,227,256,274,275,282], and the remaining 124 reported clinical accuracy.

The clinical accuracy studies were divided into 214 datasets, while the 9 analytical accuracy studies accounted for 63 datasets. A total of 61 different Ag-RDTs were evaluated (48 lateral flow with visual readout and 12 requiring an automated reader), with 56 being assessed in a clinical accuracy study. Thirty-nine studies reported data for more than 1 test, and 19 of these studies conducted a head-to-head assessment, i.e., testing at least 2 Ag-RDTs on the same sample or participant. The reference method was RT-PCR in all except 1 study, which used viral culture [281].

The most common reasons for testing were the occurrence of symptoms (55/19.9% of datasets), screening independent of symptoms (19/6.9%), and close contact with a SARS-CoV-2 confirmed case (10/3.6%). In 79 (28.6%) of the datasets, persons were tested due to more than 1 of these reasons, and for 163 datasets (59.1%), the reason for testing was unclear.

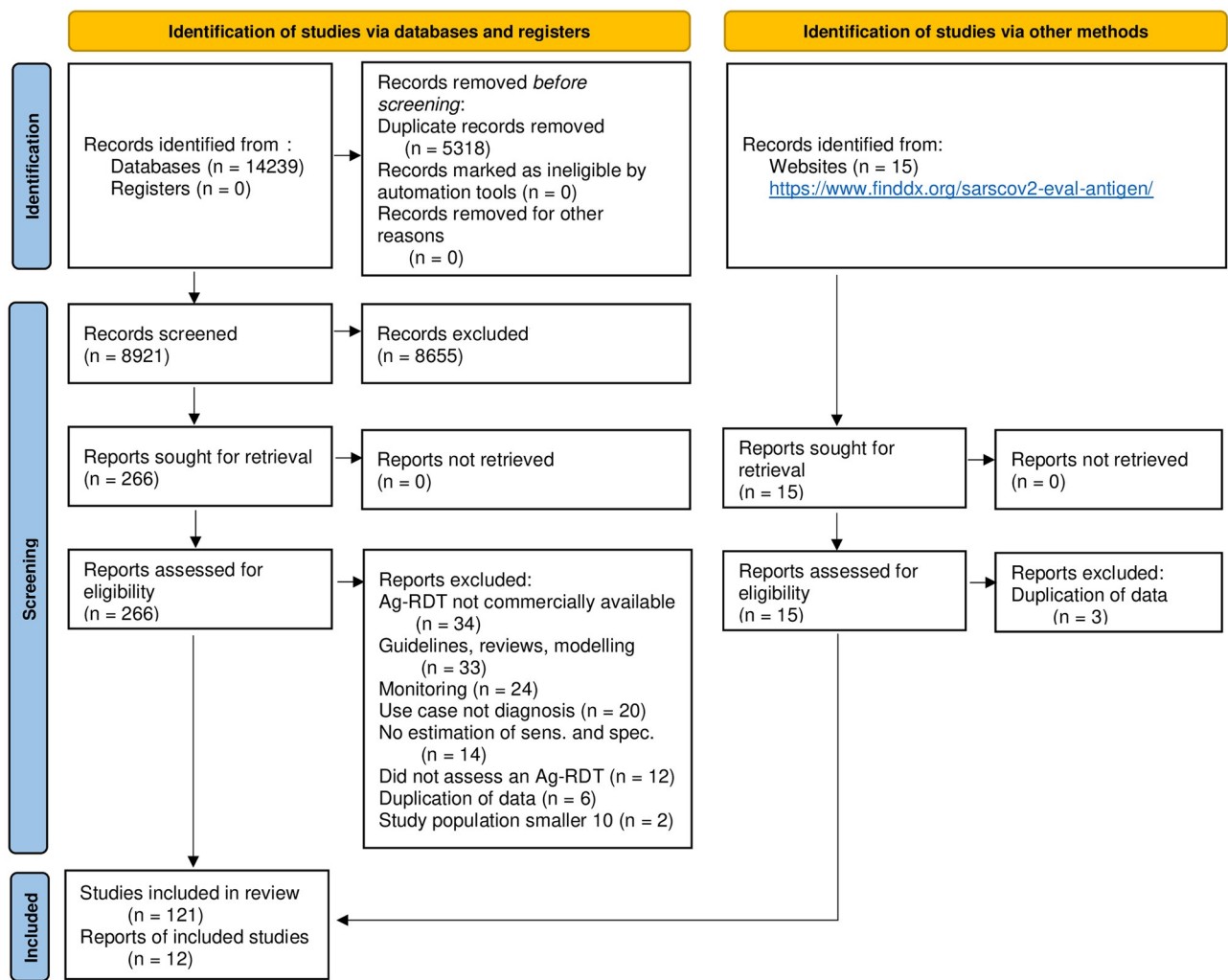

**Fig 1. PRISMA flow diagram.** Based on Page et al. [297]. Ag-RDT, antigen rapid diagnostic test; IFU, instructions for use; sens., sensitivity; spec., specificity.

In total, 113,242 Ag-RDTs were performed, 112,323 (99.2%) in clinical accuracy studies and 919 (0.8%) in analytical accuracy studies. In the clinical accuracy studies, the mean number of samples per study was 525 (range 16 to 6,954). Only 4,752 (4.2%) tests were performed on pediatric (age group <18 years) samples, and 21,351 (18.9%) on samples from adults (age group ≥18 years). For the remaining 87,139 (76.9%) samples, the age of the persons tested was not specified. Symptomatic patients comprised 36,981 (32.7%) samples; 32,799 (29.0%) samples originated from asymptomatic patients, and for 42,462 (38.4%) samples, the patient's symptom status could not be identified. The most common sample type evaluated was NP and mixed NP/OP (67,036 samples, 59.2%), followed by AN/MT (27,045 samples, 23.9%). There was substantially less testing done for the other sample types, with 6,254 (5.5%) tests done from OP samples, 1,351 (1.2%) from saliva, and 219 (0.2%) from BAL/TW, and for 11,337 (10.0%) tests, we could not identify the type of sample.

Of the datasets assessing clinical accuracy, 89 (41.6%) involved testing according to the manufacturers' recommendations (i.e., IFU-conforming), while 100 (46.7%) were not IFU-conforming, and for 25 (11.7%) it was unclear. The most common deviations from the IFU

were (1) use of samples that were prediluted in transport media not recommended by the manufacturer (80 datasets; 7 unclear), (2) use of banked samples (60 datasets; 14 unclear), and (3) use of a sample type that was not recommended for the Ag-RDT (17 datasets; 8 unclear).

A summary of the tests evaluated in clinical accuracy studies, including study identification, sample size, sample type, sample condition, and IFU conformity, can be found in Table 1. The Panbio test by Abbott (Germany; henceforth called Panbio) was reported the most frequently, with 39 (18.2%) datasets and 28,089 (25.0%) tests, while the Standard Q test by SD Biosensor (South Korea; distributed in Europe by Roche, Germany; henceforth called Standard Q) was assessed in 37 (17.3%) datasets, with 16,820 (15.0%) tests performed. Detailed results for each clinical accuracy study are available in S1 Fig.

## Methodological quality of studies

The findings on study quality using the QUADAS-2 tool are presented in Figs 2 and 3. In 190 (88.8%) datasets a relevant patient population was assessed. However, for only 44 (20.6%) of the datasets was patient selection considered representative of the setting and population chosen (i.e., they avoided inappropriate exclusions and a case–control design, and enrollment occurred consecutively or randomly).

The conduct and interpretation of the index tests was considered to have low risk for introduction of bias in 113 (52.8%) datasets (through, e.g., appropriate blinding of persons interpreting the visual readout). However, for 99 (46.3%) datasets, sufficient information to clearly judge the risk of bias was not provided. In only 89 (41.6%) datasets were the Ag-RDTs performed according to IFU, while 100 (46.7%) were not IFU-conforming, potentially impacting the diagnostic accuracy (for 25 [11.7%] datasets the IFU status was unclear).

In 81 (37.9%) datasets, the reference standard was performed before the Ag-RDT, or the operator conducting the reference standard was blinded to the Ag-RDT results, resulting in a low risk of bias. In almost all other datasets (132/61.7%), this risk could not be assessed due to missing data. The applicability of the reference test was judged to be of low concern for all datasets, as cell culture and RT-PCR are expected to adequately define the target condition.

In 209 (97.7%) datasets, the sample for the index test and reference test were obtained at the same time, while this was unclear in 5 (2.3%) datasets. All samples included in a dataset were subjected to the same type of RT-PCR in 145 (67.8%) datasets, while different types of RT-PCR were used within the same dataset in 50 (23.4%) datasets. For 19 (8.9%) datasets, it was unclear. Furthermore, for 11 (5.1%) datasets, there was a concern that not all selected patients were included in the analysis.

Finally, 32 (24.1%) of the studies received financial support from the Ag-RDT manufacturer, and in another 9 (6.8%) studies, employment of the authors by the manufacturer of the Ag-RDT studied was indicated. Overall, a competing interest was found in 33 (24.8%) of the studies.

## Detection of SARS-CoV-2 infection

Out of 214 clinical datasets (from 124 studies), 20 were excluded from the meta-analysis because they included fewer than 20 RT-PCR positive samples. A further 21 datasets were missing either sensitivity or specificity and were only considered for univariate analyses. Across the remaining 173 datasets, including any test and type of sample, the pooled sensitivity and specificity were 71.2% (95% CI 68.2% to 74.0%) and 98.9% (95% CI 98.6% to 99.1%), respectively. If testing was performed in conformity with IFU, sensitivity increased to 76.3% (95% CI 73.1% to 79.2%), while non-IFU-conforming testing had a sensitivity of 65.9% (95%

**Table 1. Clinical accuracy data for Ag-RDTs against SARS-CoV-2.**

| Reference, first author, dataset ID | Study location | Sample type | Sample condition | IFU-conforming | Sample size | Sensitivity (95% CI) | Specificity (95% CI) |
|---|---|---|---|---|---|---|---|
| **AAZ, COVID-VIRO (LFA)** | | | | | | | |
| [287] Schwob, a35.3 | Switzerland | NP | Fresh | Yes | 324 | 84.1% (76.9%, 89.7%) | 100% (98.0%*, 100%*) |
| **Abbott, BinaxNOW (LFA)** | | | | | | | |
| [224] Pollock, f17.1 | US | AN | Fresh | Yes | 2,308 | 77.4% (72.2%, 82.1%) | 99.4% (99.0%, 99.7%) |
| [283] Pilarowski, a29.1 | US | AN/MT | Fresh | Yes | 878 | 57.7% (36.9%*, 76.6%*) | 100%* (99.6%*, 100%*) |
| [197] James, f23.1 | US | AN | Fresh | Yes | 2,339 | 56.6% (48.3%*, 64.6%*) | 99.9% (99.6%, 100%) |
| [217] Okoye, f51.1 | US | MT | Fresh | Yes | 2,645 | 53.3% (37.9%*, 68.3%*) | 100% (99.9%, 100%) |
| **Abbott, Panbio (LFA)** | | | | | | | |
| [175] Domínguez Fernández, f49.1 | Spain | Unclear | Fresh | Unclear | 30 | 95.0% (75.1%*, 99.9%*) | 100% (69.2%*, 100%*) |
| [250] Alemany, a02.1 | Spain | NP | Banked | No | 919 | 93.4% (91.5%, 95.0%) | 100% (95.8%, 100%) |
| [184] FIND, f42.2 | Germany | NP | Fresh | Yes | 281 | 90.9% (78.3%*, 97.5%*) | 99.2% (97.0%, 99.9%*) |
| [276] Merino-Amador, a25.1 | Spain | NP | Fresh | Yes | 958 | 90.5% (87.0%*, 93.4%*) | 98.8% (97.6%*, 99.5%*) |
| [267] Krüger, a52.1 | Germany | NP | Fresh | Yes | 1,034 | 87.5% (79.6%*, 93.2%*) | 99.9% (99.4%, 100%) |
| [287] Schwob, a35.2 | Switzerland | NP | Fresh | Yes | 271 | 86.1% (78.6%, 91.7%) | 100% (97.6%*, 100%*) |
| [235] Stokes, f65.1 | Canada | NP | Fresh | Yes | 1,641 | 86.2%* (81.5%*, 90.1%*) | 99.9% (99.5%, 100%) |
| [252] Berger, a05.1 | Switzerland | NP | Fresh | Yes | 535 | 85.5% (78.0%, 91.2%) | 100% (99.1%, 100%) |
| [177] Faíco-Filho, f63.1 | Brazil | NP | Fresh | Yes | 127 | 84.3% (73.6%*, 91.9%*) | 98.2%* (90.6%*, 100%*) |
| [196] Jääskeläinen, f50.3 | Finland | NP | Banked | No | 190 | 82.9%* (76.0%*, 88.5%*) | 100% (90.7%*, 100%*) |
| [247] Abdulrahman, a01.1 | Bahrain | AN/MT | Fresh | No | 4,183 | 82.1% (79.2%, 84.8%) | 99.1% (98.8%, 99.4%) |
| [263] Gremmels, a12.2 | Netherlands | NP | Fresh | Yes | 208 | 81.0% (69.1%*, 89.8%*) | 100% (97.5%, 100%) |
| [214] Ngo Nsoga, f28.1 | Switzerland | OP | Fresh | No | 402 | 81.0% (74.2%, 86.6%) | 99.1% (96.9%, 99.9%) |
| [245] Yin, f82.2 | Belgium | NP | Fresh | Yes | 101 | 80.8% (68.1%, 89.2%) | Not provided |
| [249] Albert, a03.1 | Spain | NP | Fresh | Yes | 412 | 79.6% (66.5%*, 89.4%*) | 100% (99.0%, 100%) |
| [250] Alemany, a02.2 | Spain | AN/MT | Banked | No | 487 | 79.5% (71.0%, 86.4%) | 98.6%* (96.9%, 99.6%) |
| [258] Fenollar, a11.1 | France | NP | Fresh | Yes | 341 | 75.5% (69.0%*, 81.2%*) | 94.9% (89.8%*, 97.9%*) |
| [270] Linares, a20.1 | Spain | NP | Fresh | Unclear | 255 | 73.3% (60.3%*, 83.9%*) | 100% (98.1%*, 100%*) |
| [263] Gremmels, a12.1 | Netherlands | NP | Fresh | Yes | 1,367 | 72.7%* (64.5%, 79.9%) | 100% (99.7%, 100%) |
| [192] Halfon, f18.1 | France | NP | Unclear | No | 200 | 72.0% (62.1%*, 80.5%*) | 99.0% (94.6%*, 100%) |
| [253] Bulilete, a07.1 | Spain | NP | Fresh | Yes | 1,362* | 71.4% (63.2%*, 78.7%) | 99.8% (99.4%, 99.9%) |
| [165] Akingba, f30.1 | South Africa | NP | Fresh | Unclear | 657* | 69.7%* (61.5%*, 77.0%*) | 99.4%* (98.3%*, 99.9%*) |
| [174] Del Vecchio, f66.1 | Italy | Unclear | Fresh | Unclear | 1,441 | 68.9% (55.7%, 80.1%) | 99.9% (99.6%, 100%) |
| [178] Favresse, f31.2 | Belgium | NP | Fresh | No | 188 | 67.7% (57.4%, 76.9%) | 100% (96.1%, 100%) |
| [257] Drevinek, a10.1 | Czech Republic | NP | Fresh | Yes | 591 | 66.4% (59.8%*, 72.5%*) | 100% (99.0%, 100%) |
| [205] L'Huillier, f72.1 | Switzerland | NP | Fresh | Yes | 822 | 65.5%* (56.3%*, 74.0%*) | 99.9%* (99.2%*, 100%) |
| [221] Pérez-García, f52.2 | Spain | NP | Banked | No | 320 | 60.0% (52.2%, 67.4%) | 100% (97.6%, 100%) |
| [248] Agulló, a56.1 | Spain | NP | Fresh | Yes | 652* | 57.6%* (48.7%*, 66.1%*) | 99.8% (98.9%*, 100%) |
| [267] Krüger, a52.2 | Germany | OP | Fresh | No | 74 | 50.0% (1.3%, 98.7%) | 100% (94.9%, 100%) |
| [286] Schildgen, a33.2 | Germany | BAL/TW | Unclear | No | 73 | 50.0% (34.2%*, 65.8%*) | 77.4% (58.9%*, 90.4%*) |
| [292] Torres, a37.1 | Spain | NP | Fresh | Yes | 634 | 48.1% (36.7%*, 59.6%*) | 100% (99.3%, 100%) |
| [244] Wagenhäuser, f89.2 | Germany | OP | Fresh | No | 1,029 | 46.7% (24.8%, 69.9%) | 99.6% (99.0%, 99.9%) |
| [243] Villaverde, f55.1 | Spain | NP | Fresh | Yes | 1,620 | 45.4% (34.1%, 57.2%) | 99.8% (99.4%, 99.9%) |
| [248] Agulló, a56.2 | Spain | AN/MT | Fresh | No | 659 | 44.7% (36.1%, 53.6%) | 100% (99.3%*, 100%) |
| [279] Olearo, a54.2 | Germany | OP | Unclear | No | 184 | 44.0% (33.2%*, 55.3%*) | 100% (96.4%*, 100%) |

*(Continued)*

**Table 1.** (Continued)

| Reference, first author, dataset ID | Study location | Sample type | Sample condition | IFU-conforming | Sample size | Sensitivity (95% CI) | Specificity (95% CI) |
|---|---|---|---|---|---|---|---|
| [170] Caruana, f34.2 | Switzerland | NP | Fresh | No | 532 | 41.2% (32.1%*, 50.8%*) | 99.5% (98.3%*, 99.9%*) |
| [167] Baro, f33.1 | Spain | NP | Banked | No | 286 | 38.6% (29.1%, 48.8%) | 99.5% (97.0%, 100%) |
| [248] Agulló, a56.3 | Spain | Saliva | Fresh | No | 610 | 23.1% (16.0%, 31.7%*) | 100% (99.2%*, 100%) |
| [213] Muhi, f90.1 | Australia | NP | Fresh | Yes | 2,413 | Not provided | 100% (99.7%, 100%) |
| **Abbott, Panbio (nasal sampling) (LFA)** | | | | | | | |
| [184] FIND, f42.1 | Germany | AN/MT | Fresh | Yes | 281 | 86.4% (72.6%*, 94.8%*) | 99.2% (97.0%, 99.9%*) |
| **Access Bio, CareStart COVID-19 Antigen Test (LFA)** | | | | | | | |
| [225] Pollock, f59.1 | US | AN | Fresh | Yes | 1,498 | 57.7% (51.1%, 64.1%) | 98.3% (97.5%, 99.0%) |
| **Assure Tech, Ecotest COVID-19 Antigen Rapid Test (LFA)** | | | | | | | |
| [194] Homza, f87.1 | Czech Republic | NP | Fresh | Yes | 318 | 75.7% (66.5%, 83.5%) | 96.7% (93.3%, 98.7%) |
| **Becton, Dickinson and Company, BD Veritor (requires reader)** | | | | | | | |
| [281] Pekosz, a28.1 | US | NP | Fresh | No | 251 | 96.4% (81.7%*, 99.9%*) | 98.7% (96.1%, 99.7%) |
| [293] Van der Moeren, a39.1 | Netherlands | MT/OP | Banked | No | 351* | 94.1% (71.1%, 100%) | 100% (98.9%, 100%) |
| [190] Gomez Marti, f46.2 | US | AN | Fresh | Unclear | Unknown | 93.8% (79.2%*, 99.2%*) | Not provided |
| [245] Yin, f82.1 | Belgium | NP | Fresh | Yes | 177 | 87.7% (80.0%, 92.7%) | Not provided |
| [296] Young, a43.1 | US | NP | Banked | No | 251 | 76.3%* (59.8%*, 88.6%*) | 99.5%* (97.4%*, 99.9%*) |
| [202] Kilic, f71.1 | US | AN | Fresh | Yes | 1,384 | 66.4% (57.0%, 74.9%) | 98.8% (98.1%, 99.3%) |
| [231] Schuit, f64.1 | Netherlands | NP | Fresh | No | 2,678 | 63.9% (57.4%, 70.1%) | 99.6% (99.3%, 99.8%) |
| [170] Caruana, f34.4 | Switzerland | NP | Fresh | No | 532 | 41.2% (32.1%*, 50.8%*) | 99.8%* (98.7%*, 100%*) |
| **Becton, Dickinson and Company, Hometest (LFA)** | | | | | | | |
| [234] Stohr, f45.1 | Netherlands | AN | Fresh | Unclear | 1,604 | 48.9% (41.3%*, 56.5%*) | 99.9% (99.5%, 100%) |
| **Beijing Savant Biotechnology, SARS-CoV-2 detection kit (LFA)** | | | | | | | |
| [295] Weitzel, a41.3 | Chile | NP/OP | Banked | No | 109 | 16.7% (9.2%*, 26.8%*) | 100% (88.8%*, 100%) |
| **Biotime, COVID-19 Antigen Test Cassette (LFA)** | | | | | | | |
| [232] Seitz, f68.1 | Austria | Saliva | Fresh | Yes | 40 | 44.4% (21.5%*, 69.2%*) | 100% (84.6%*, 100%*) |
| **Bionote, NowCheck (LFA)** | | | | | | | |
| [185] FIND, f91.1 | Brazil | AN | Fresh | Yes | 218 | 89.9% (81.0%*, 95.5%*) | 98.6% (94.9%, 99.8%*) |
| [185] FIND, f91.2 | Brazil | NP | Fresh | Yes | 218 | 89.9% (81.0%*, 95.5%*) | 98.6% (94.9%, 99.8%*) |
| [259] FIND, a61.1 | Brazil | NP | Fresh | Yes | 400 | 89.2% (81.5%*, 94.5%*) | 97.3% (94.8%, 98.8%*) |
| [228] Rottenstreich, f53.1 | Israel | NP | Unclear | Unclear | 1,326 | 55.6% (21.2%, 86.3%) | 100% (99.7%, 100%) |
| **Biotical Health, SARS-CoV-2 Ag Card (LFA)** | | | | | | | |
| [178] Favresse, f31.1 | Belgium | NP | Fresh | No | 188 | 66.7% (56.3%, 76.0%) | 98.9% (94.1%, 99.9%) |
| **Boditech Medical, iChroma COVID-19 Ag Test (requires reader)** | | | | | | | |
| [181] FIND, f39.1 | Switzerland | NP | Fresh | Yes | 232 | 73.2% (57.1%*, 85.8%*) | 100% (98.0%, 100%) |
| **CerTest Biotec, SARS-CoV-2 one step test card (LFA)** | | | | | | | |
| [221] Pérez-García, f52.1 | Spain | NP | Banked | No | 320 | 53.5% (45.7%, 61.2%) | 100% (97.6%, 100%) |
| **Coris BioConcept, COVID-19 Ag Respi-Strip (LFA)** | | | | | | | |
| [245] Yin, f82.3 | Belgium | NP | Fresh | Yes | 135 | 80.0% (69.2%, 87.7%) | Not provided |
| [277] Mertens, a48.1 | Belgium | NP | Banked | No | 328 | 57.6% (48.7%*, 66.1%*) | 99.5% (97.2%*, 100%*) |
| [269] Lambert-Niclot, a18.1 | France | NP | Fresh | No | 138 | 50.0% (39.5%, 60.5%) | 100% (92.0%*, 100%) |
| [4] Krüger, a17.3 | Germany/England | NP/OP | Unclear | No | 417 | 50.0% (21.5%*, 78.5%*) | 95.8% (93.4%, 97.4%) |
| [172] Ciotti, f24.1 | Italy | NP | Fresh | Unclear | 50 | 30.8% (17.0%, 47.6%) | 100% (71.5%, 100%) |
| [288] Scohy, a34.1 | Belgium | NP | Fresh | No | 148 | 30.2% (21.7%, 39.9%) | 100% (91.6%*, 100%*) |
| [294] Veyrenche, a40.1 | France | NP | Fresh | No | 65 | 28.9%* (16.4%*, 44.3%*) | 100% (83.2%, 100%) |

(Continued)

**Table 1.** (Continued)

| Reference, first author, dataset ID | Study location | Sample type | Sample condition | IFU-conforming | Sample size | Sensitivity (95% CI) | Specificity (95% CI) |
|---|---|---|---|---|---|---|---|
| **Denka, Quick Navi (LFA)** | | | | | | | |
| [237] Takeuchi, f12.1 | Japan | NP | Fresh | Unclear | 1,186 | 86.7% (78.6%, 92.5%) | 100% (99.7%, 100%) |
| [238] Takeuchi, f60.1 | Japan | AN | Fresh | Unclear | 862 | 72.5% (58.3%, 84.1%) | 100% (99.5%*, 100%) |
| **DiaSorin, LIAISON SARS-CoV-2 Ag (LFA)** | | | | | | | |
| [206] Lefever, f70.1 | Belgium | NP | Banked | No | 414 | 67.6%* (60.8%*, 74.0%*) | 100% (98.3%*, 100%) |
| **Dräger, Antigen Test SARS-CoV-2 (LFA)** | | | | | | | |
| [218] Osmanodja, f79.1 | Germany | NP/OP | Fresh | Yes | 379 | 88.6% (78.7%, 94.9%) | 99.7% (98.2%, 100%) |
| **E25Bio, Rapid Diagnostic Test (LFA)** | | | | | | | |
| [223] Pickering, f73.2 | UK | AN/OP | Banked | No | 200 | 75.0% (65.3%*, 83.1%*) | 86% (77.6%*, 92.1%*) |
| **ECO Diagnóstica, COVID-19 Ag (LFA)** | | | | | | | |
| [180] Filgueiras, f14.1 | Brazil | NP | Fresh | Unclear | 150 | 69.1% (55.2%*, 80.9%*) | 98.8% (93.5%*, 100%) |
| **Fujirebio, ESPLINE SARS-CoV-2 (LFA)** | | | | | | | |
| [290] Takeda, a50.1 | Japan | NP | Unclear | No | 162 | 80.6%* (68.6%*, 89.6%*) | 100%* (96.4%*, 100%*) |
| [186] FIND, f92.1 | Germany | NP/OP | Fresh | No | 723 | 78.6% (69.8%*, 85.8%*) | 100% (99.4%, 100%) |
| [230] Sberna, f83.1 | Italy | Saliva | Unclear | Unclear | 136 | 8.1% (2.7%, 17.8%) | 100% (95.1%, 100%) |
| **Fujirebio, Lumipulse G SARS-CoV-2 Ag (requires reader)** | | | | | | | |
| [189] Gili, f57.2 | Italy | NP | Banked | No | 226 | 100% (96.0%*, 100%*) | 92.1% (90.7%*, 93.4%*) |
| [189] Gili, f57.1 | Italy | NP | Fresh | No | 1,738 | 90.5% (82.8%*, 95.6%*) | 91.6% (85.5%*, 95.7%*) |
| [193] Hirotsu, f47.1 | Japan | NP | Banked | No | 1,033 | 92.5% (79.6%*, 98.4%*) | 100%* (99.6%*, 100%*) |
| [168] Basso, f10.1 | Italy | NP | Fresh | Yes | 234 | 81.6% (71.9%*, 89.1%*) | 93.9%* (88.7%*, 97.2%*) |
| [166] Asai, f74.1 | Japan | Saliva | Unclear | Yes | 305 | 77.8% (65.5%*, 87.3%*) | 98.3% (95.8%*, 99.5%*) |
| [168] Basso, f10.2 | Italy | Saliva | Fresh | Yes | 223 | 41.3% (30.4%, 52.8%) | 98.6% (95.0%, 99.8%) |
| **Guangzhou Wondfo Biotech, 2019-nCoV Antigen Test (LFA)** | | | | | | | |
| [183] FIND, f41.1 | Switzerland | NP | Fresh | Yes | 328 | 85.7% (73.8%*, 93.6%*) | 100% (98.7%*, 100%*) |
| **Humasis, COVID-19 Ag Test (LFA)** | | | | | | | |
| [169] Bruzzone, f86.2 | Italy | Unclear | Banked | No | 21 | 85.7% (63.7%*, 97%*) | Not provided |
| **Healgen, Rapid COVID-19 Ag Test (LFA)** | | | | | | | |
| [178] Favresse, f31.3 | Belgium | NP | Fresh | No | 188 | 77.1% (67.4%, 85.1%) | 96.7% (90.8%, 99.3%) |
| **Innova Medical Group, INNOVA SARS-CoV-2 Antigen Rapid Qualitative Test (LFA)** | | | | | | | |
| [223] Pickering, f73.1 | UK | AN/OP | Banked | No | 200 | 89.0% (81.2%*, 94.4%*) | 99.0% (94.6%, 100%) |
| [195] Houston, f25.1 | UK | NP | Fresh | Yes | 242 | 86.4% (81.9%*, 90.2%*) | 95.1% (92.7%*, 96.9%*) |
| [223] Pickering, f73.10 | UK | AN/OP | Banked | No | 23 | 82.6% (61.2%*, 95.0%*) | Not provided |
| [223] Pickering, f73.11 | UK | AN/OP | Banked | No | 23 | 82.6% (61.2%*, 95.0%*) | Not provided |
| [222] Peto, f21.1 | UK | Unclear | Unclear | Unclear | 6,954 | Not provided | 99.7% (99.5%*, 99.8%*) |
| [222] Peto, f21.4 | UK | Unclear | Unclear | Unclear | 198 | 78.8% (72.4%, 84.3%) | Not provided |
| [223] Pickering, f73.12 | UK | AN/OP | Banked | No | 23 | 78.3% (56.3%*, 92.5%*) | Not provided |
| [223] Pickering, f73.8 | UK | AN/OP | Banked | No | 110 | 78.2% (69.3%*, 85.5%*) | Not provided |
| [222] Peto, f21.3 | UK | Unclear | Unclear | Unclear | 223 | 70.0% (63.5%, 75.9%) | Not provided |
| [246] Young, f56.1 | UK | NP | Fresh | Unclear | 803 | 62.1%* (55.3%*, 68.7%*) | 100% (99.4%, 100%) |
| [222] Peto, f21.2 | UK | Unclear | Unclear | Unclear | 372 | 57.5% (52.3%, 62.6%) | Not provided |
| [179] Ferguson, f85.1 | UK | AN | Fresh | Yes | 720 | 3.2% (0.6%, 15.6%) | 100% (99.5%, 100%) |
| **JOYSBIO Biotechnology, COVID-19 Antigen Rapid Test Kit (LFA)** | | | | | | | |
| [182] FIND, f40.1 | Switzerland | NP | Fresh | Yes | 265 | 70.5% (54.8%*, 83.2%*) | 99.1% (96.8%*, 99.9%*) |
| [194] Homza, f87.2 | Czech Republic | NP | Fresh | Yes | 225 | 57.8% (46.9%, 68.1%) | 98.5% (94.8%, 99.8%) |
| **Lab Care Diagnostics, PathoCatch/ACCUCARE SARS-CoV-2 Antigen Test (LFA)** | | | | | | | |
| [239] Thakur, f88.1 | India | NP | Fresh | Yes | 677 | 34.5% (24.5%, 45.6%) | 99.8% (99.1%, 100%) |

(Continued)

**Table 1.** (Continued)

| Reference, first author, dataset ID | Study location | Sample type | Sample condition | IFU-conforming | Sample size | Sensitivity (95% CI) | Specificity (95% CI) |
|---|---|---|---|---|---|---|---|
| **Lepu Medical Technology, SARS-CoV-2 Antigen Rapid Test Kit (LFA)** | | | | | | | |
| [167] Baro, f33.4 | Spain | NP | Banked | No | 286 | 45.5% (35.6%, 55.8%) | 89.2% (83.8%, 93.3%) |
| **Liming Bio, SARS-CoV-2 Ag-RDT (LFA)** | | | | | | | |
| [295] Weitzel, a41.2 | Chile | NP/OP | Banked | No | 19 | 0% (0%, 29.9%) | 90.0% (59.6%, 98.2%) |
| **LumiraDx, COVID-19 SARS-CoV-2 Antigen Test (requires reader)** | | | | | | | |
| [176] Drain, f43.1 | UK/US | AN | Fresh | Yes | 257 | 97.6% (91.6%*, 99.7%*) | 96.6% (92.6%*, 98.7%*) |
| [176] Drain, f43.2 | UK/US | NP | Fresh | Yes | 255 | 97.5% (86.8%*, 99.9%*) | 97.7% (94.7%, 99.2%*) |
| [204] Krüger, f58.1 | Germany | MT | Fresh | Yes | 761 | 82.2% (75.0%*, 88.0%*) | 99.3% (98.3%, 99.7%) |
| [169] Bruzzone, f86.6 | Italy | Unclear | Banked | No | 23 | 69.6% (47.1%*, 86.8%*) | Not provided |
| [203] Kohmer, f32.4 | Germany | NP | Fresh | No | 100 | 50.0% (38.1%, 61.9%) | 100% (86.8%, 100%) |
| [211] Micocci, f77.1 | UK | NP | Fresh | Unclear | 241 | 75.0%* (34.9%*, 96.8%*) | 96.1%* (92.7%*, 98.2%*) |
| **MEDsan, SARS-CoV-2 Antigen Rapid Test (LFA)** | | | | | | | |
| [279] Olearo, a54.3 | Germany | OP | Unclear | No | 184 | 45.2%* (34.3%*, 56.5%) | 97.0% (91.5%, 99.4%*) |
| [244] Wagenhäuser, f89.3 | Germany | OP | Fresh | Yes | 3,221 | 36.5% (24.7%*, 49.6%*) | 99.6% (99.3%, 99.8%) |
| **Mologic, COVID-19 Rapid Antigen Test (LFA)** | | | | | | | |
| [187] FIND, f93.1 | Germany | AN/MT | Fresh | Yes | 665 | 90.7% (85.7%*, 94.4%*) | 100% (99.2%, 100%) |
| **nal von minden, NADAL (LFA)** | | | | | | | |
| [188] FIND, f94.1 | Switzerland | NP | Fresh | Yes | 462 | 88.4% (78.4%, 94.9%*) | 99.2% (97.8%, 99.7%) |
| [236] Strömer, f11.1 | Germany | NP | Banked | No | 124 | 63.7%* (54.6%*, 72.2%*) | Not provided |
| [244] Wagenhäuser, f89.1 | Germany | OP | Fresh | Yes | 806 | 56.5% (34.5%*, 76.8%*) | 100% (99.5%, 100%) |
| [203] Kohmer, f32.3 | Germany | NP | Fresh | No | 100 | 24.3% (15.1%, 35.7%) | 100% (86.8%, 100%) |
| **NanoEntek, FREND COVID-19 Ag (requires reader)** | | | | | | | |
| [169] Bruzzone, f86.7 | Italy | Unclear | Banked | No | 60 | 93.3% (83.8%*, 98.2%*) | Not provided |
| **NDFOS, ND COVID-19 Ag Test (LFA)** | | | | | | | |
| [194] Homza, f87.3 | Czech Republic | NP | Fresh | Yes | 191 | 70.1% (58.6%, 80.0%) | 56.1% (46.4%, 65.4%) |
| **Ortho Clinical Diagnostics, VITROS SARS-CoV-2 Antigen Test (requires reader)** | | | | | | | |
| [178] Favresse, f31.5 | Belgium | NP | Fresh | No | 188 | 83.3% (74.4%, 90.2%) | 100% (96.1%, 100%) |
| **Precision Biosensor, Exdia COVID-19 Ag (requires reader)** | | | | | | | |
| [170] Caruana, f34.3 | Switzerland | NP | Fresh | No | 532 | 48.3% (38.8%, 57.8%*) | 99.5% (98.3%*, 99.9%*) |
| **PRIMA Lab, COVID-19 Antigen Rapid Test (LFA)** | | | | | | | |
| [169] Bruzzone, f86.3 | Italy | Unclear | Banked | No | 50 | 66.0% (51.2%*, 78.8%*) | Not provided |
| **Quidel, Sofia SARS Antigen FIA (requires reader)** | | | | | | | |
| [284] Porte, a32.1 | Chile | NP/OP | Banked | No | 64 | 93.8% (79.2%*, 99.2%*) | 96.9% (83.8%*, 99.9%*) |
| [196] Jääskeläinen, f50.1 | Finland | NP | Banked | No | 188 | 80.4% (73.1%*, 86.5%*) | 100% (91.2%*, 100%*) |
| [251] Beck, a04.1 | US | NP | Fresh | Yes | 346 | 77.0% (64.5%*, 86.8%*) | 99.6% (98.1%*, 100%*) |
| [265] Herrera, a46.1 | US | Unclear | Unclear | Unclear | 1,172 | 76.8% (72.6%, 80.5%) | 99.2% (98.2%, 99.7%) |
| [190] Gomez Marti, f46.1 | US | MT | Fresh | Unclear | 427 | 72.0% (56.3%, 84.7%*) | 99.7%* (98.6%*, 100%*) |
| **RapiGEN, Biocredit Covid-19 Ag (LFA)** | | | | | | | |
| [289] Shrestha, a36.1 | Nepal | NP | Fresh | Yes | 113 | 85.0% (71.7%, 93.8%*) | 100% (94.6%*, 100%*) |
| [260] FIND, a62.1 | Brazil | NP | Fresh | Yes | 476 | 74.4% (65.5%, 82.0%*) | 98.9%* (97.2%, 99.7%*) |
| [295] Weitzel, a41.1 | Chile | NP/OP | Banked | No | 109 | 62.0% (50.4%, 72.7%*) | 100% (88.4%*, 100%) |
| [233] Shidlovskaya, f61.1 | Russia | NP | Fresh | Yes | 106 | 56.4% (44.7%, 67.6%) | 100% (87.7%, 100%) |
| [260] FIND, a62.2 | Germany | NP | Fresh | Yes | 1,239 | 52.0% (31.3%*, 72.2%*) | 100% (99.7%, 100%) |
| [169] Bruzzone, f86.4 | Italy | Unclear | Banked | No | 23 | 39.1% (19.7%*, 61.5%*) | Not provided |
| [286] Schildgen, a33.1 | Germany | BAL/TW | Unclear | No | 73 | 33.3% (19.6%*, 49.6%*) | 87.1% (70.2%*, 96.4%*) |
| [200] Kenyeres, f84.1 | Hungary | NP | Fresh | No | 37 | 8.1% (1.7%*, 21.9%*) | Not provided |

*(Continued)*

**Table 1.** (Continued)

| Reference, first author, dataset ID | Study location | Sample type | Sample condition | IFU-conforming | Sample size | Sensitivity (95% CI) | Specificity (95% CI) |
|---|---|---|---|---|---|---|---|
| **R-Biopharm, RIDA QUICK SARS-CoV-2 Antigen (LFA)** | | | | | | | |
| [291] Toptan, a55.1 | Germany | NP/OP | Banked | No | 67 | 77.6% (64.7%*, 87.5%*) | 100% (66.4%*, 100%*) |
| [291] Toptan, a55.2 | Germany | Unclear | Banked | No | 70 | 50.0% (31.9%*, 68.1%*) | 100% (90.8%*, 100%*) |
| [203] Kohmer, f32.1 | Germany | NP | Fresh | No | 100 | 39.2% (28.0%, 51.2%) | 96.2% (80.4%, 99.9%) |
| **Roche, Elecsys SARS-CoV-2 Antigen Test (requires reader)** | | | | | | | |
| [216] Nörz, f78.1 | Germany | NP/OP | Banked | No | 3,139 | 60.2% (55.2%, 65.1%) | 99.9% (99.6%, 100%) |
| **Roche, SARS-CoV-2 Rapid Antigen Test (LFA)** | | | | | | | |
| [240] Thell, f81.1 | Austria | Unclear | Fresh | Unclear | 591 | 80.3% (74.3%, 85.4%) | 99.1% (97.4%, 99.8%) |
| **Salofa Oy, Sienna COVID-19 Antigen Rapid Test Cassette (LFA)** | | | | | | | |
| [209] Mboumba Bouassa, f67.1 | France | NP | Banked | No | 100 | 90.0% (82.4%*, 95.1%*) | 100% (92.9%*, 100%) |
| **SD Biosensor, Standard F (requires reader)** | | | | | | | |
| [284] Porte, a32.2 | Chile | NP/OP | Banked | No | 64 | 90.6% (75.0%*, 98.0%*) | 96.9% (83.8%*, 99.9%*) |
| [169] Bruzzone, f86.5 | Italy | Unclear | Banked | No | 60 | 86.7% (75.4%*, 94.1%*) | Not provided |
| [261] FIND, a63.1 | Brazil | NP | Fresh | Yes | 453 | 77.5% (69.0%*, 84.6%*) | 97.9% (95.7%, 99.2%*) |
| [261] FIND, a63.2 | Germany | NP | Fresh | Yes | 676 | 69.2% (52.4%*, 83.0%*) | 96.9% (95.2%, 98.0%) |
| [257] Drevinek, a10.2 | Czech Republic | NP | Fresh | Yes | 591 | 62.3% (55.6%, 68.7%*) | 99.5% (98.0%, 99.9%) |
| [219] Osterman, f20.1 | Germany | NP/OP | Unclear | No | 360 | 60.9% (53.5%, 67.8%*) | 97.8% (95.7%, 99.0%*) |
| [273] Liotti, a22.1 | Italy | NP | Banked | No | 359 | 47.1% (37.1%, 57.1%) | 98.4% (96.0%, 99.6%) |
| **SD Biosensor/Roche, Standard Q (LFA)** | | | | | | | |
| [255] Chaimayo, a57.1 | Thailand | NP/OP | Banked | No | 454 | 98.3% (91.1%, 100%) | 98.7% (97.1%, 99.6%) |
| [169] Bruzzone, f86.1 | Italy | Unclear | Banked | No | 16 | 93.8% (71.7%, 98.9%) | Not provided |
| [201] Kernéis, f69.1 | France | NP | Fresh | Unclear | 1,109 | 94.2%* (87.0%*, 98.1%*) | 99.0% (98.2%*, 99.5%*) |
| [287] Schwob, a35.1 | Switzerland | NP | Fresh | Yes | 333 | 92.9% (86.4%, 96.9%) | 100% (98.3%*, 100%*) |
| [215] Nikolai, f35.3 | Germany | NP | Fresh | Yes | 96 | 91.2% (76.3%*, 98.1%*) | 100% (94.2%, 100%) |
| [252] Berger, a05.2 | Switzerland | NP | Fresh | Yes | 529 | 89.0% (83.7%, 93.1%) | 99.7% (98.4%, 100%) |
| [262] FIND, a64.1 | Brazil | NP | Fresh | Yes | 400 | 88.7% (81.1%*, 94.0%*) | 97.6% (95.2%, 99.0%*) |
| [286] Schildgen, a33.3 | Germany | BAL/TW | Unclear | No | 73 | 88.1% (74.4%*, 96.0%*) | 19.4% (7.5%*, 37.5%*) |
| [207] Lindner, f15.1 | Germany | NP | Fresh | Yes | 139 | 85.0% (70.2%*, 94.3%*) | 99.1% (94.9%*, 100%*) |
| [266] Igli, a15.1 | Netherlands | NP | Fresh | Yes | 970 | 84.9% (79.0%*, 89.8%*) | 99.5% (98.7%*, 99.9%*) |
| [264] Gupta, a13.1 | India | NP | Fresh | Yes | 330 | 81.8% (71.4%*, 89.7%*) | 99.6% (97.8%, 99.9%) |
| [196] Jääskeläinen, f50.2 | Finland | NP | Banked | No | 198 | 81.0% (74.0%, 86.8%*) | 100% (91.2%*, 100%*) |
| [242] Turcato, f09.1 | Italy | NP | Fresh | Unclear | 3,410 | 80.3% (74.4%*, 85.3%*) | 99.1% (98.7%*, 99.4%*) |
| [272] Lindner, a21.2 | Germany | NP | Fresh | Yes | 289 | 79.5% (63.5%*, 90.7%*) | 99.6% (97.8%, 100%) |
| [245] Yin, f82.4 | Belgium | NP | Fresh | Yes | 65 | 78.3% (58.1%, 90.3%) | Not provided |
| [4] Krüger, a17.1 | Germany/England | NP/OP | Unclear | No | 1,263 | 76.6% (62.0%*, 87.7%*) | 99.3% (98.6%*, 99.7%*) |
| [212] Möckel, f19.1 | Germany | NP/OP | Fresh | Yes | 271 | 75.3% (65.0%*, 83.8%*) | 100% (98.0%*, 100%) |
| [272] Lindner, a21.1 | Germany | AN/MT | Fresh | No | 289 | 74.4% (57.9%*87.0%*) | 99.2% (97.1%, 99.9%*) |
| [271] Lindner, a53.1 | Germany | NP | Fresh | Yes | 180 | 73.2%* (57.1%*, 85.8%*) | 99.3% (96.0%, 100%) |
| [229] Salvagno, f54.1 | Italy | NP | Unclear | No | 321 | 72.5% (64.6%, 79.5%) | 99.4% (96.8%, 100%) |
| [254] Cerutti, a08.1 | Italy | NP | Unclear | No | 185 | 72.1% (62.5%*, 80.5%*) | 100% (95.6%*, 100%*) |
| [212] Möckel, f19.2 | Germany | NP/OP | Fresh | Yes | 2,020 | 72.0% (50.6%*, 87.9%*) | 99.4% (96.9%*, 100%*) |
| [268] Krüttgen, a16.1 | Germany | NP | Banked | No | 150 | 70.7% (59.0%, 80.6%*) | 96.0% (88.8%*, 99.2%*) |
| [278] Nalumansi, a27.1 | Uganda | NP | Fresh | Yes | 262 | 70.0% (59.4%, 79.2%*) | 92.4%* (87.4%*, 95.9%*) |
| [220] Pena, f36.1 | Chile | NP | Fresh | Yes | 842 | 69.9% (58.0%*, 80.1%*) | 99.6% (98.9%, 99.9%) |
| [178] Favresse, f31.4 | Belgium | NP | Fresh | No | 188 | 69.8% (59.6%, 78.8%) | 100% (96.1%, 100%) |

*(Continued)*

**Table 1.** (Continued)

| Reference, first author, dataset ID | Study location | Sample type | Sample condition | IFU-conforming | Sample size | Sensitivity (95% CI) | Specificity (95% CI) |
|---|---|---|---|---|---|---|---|
| [219] Osterman, f20.2 | Germany | NP/OP | Unclear | No | 386 | 64.5% (58.3%*, 70.3%*) | 97.7% (95.6%, 98.9%*) |
| [194] Homza, f87.4 | Czech Republic | NP | Fresh | Yes | 139 | 61.9% (45.6%, 76.4%) | 99.0% (94.4%, 100%) |
| [231] Schuit, f64.2 | Netherlands | NP | Fresh | Yes | 1,596 | 62.9% (54.0%, 71.1%) | 99.5% (98.9%, 99.8%) |
| [226] Ristić, f44.1 | Serbia | NP | Fresh | Unclear | 120 | 58.1% (42.1%, 73.0%) | 100% (95.3%*, 100%) |
| [199] Kannian, f26.1 | India | Saliva | Unclear | No | 37 | 55.6%* (35.3%*, 74.5%*) | 100% (69.2%*, 100%*) |
| [279] Olearo, a54.1 | Germany | OP | Unclear | No | 184 | 48.8%* (37.7%*, 60.0%*) | 100% (96.4%*, 100%) |
| [167] Baro, f33.3 | Spain | NP | Banked | No | 286 | 43.6% (33.7%, 53.8%) | 96.2% (92.4%, 98.5%) |
| [203] Kohmer, f32.2 | Germany | NP | Fresh | No | 100 | 43.2% (31.8%*, 55.3%) | 100% (86.8%, 100%) |
| [170] Caruana, f34.1 | Switzerland | NP | Fresh | No | 532 | 41.2% (32.1%*, 50.8%*) | 99.8%* (98.7%*, 100%*) |
| [254] Cerutti, a08.2 | Italy | NP | Fresh | No | 145 | 40.0% (5.3%*, 85.3%*) | 100% (97.4%*, 100%*) |
| [171] Caruana, f75.1 | Switzerland | NP | Fresh | Unclear | 116 | 28.6% (3.7%*, 71.0%*) | 98.2% (93.5%*, 99.8%*) |
| **SD Biosensor/Roche, Standard Q (nasal sampling) (LFA)** | | | | | | | |
| [215] Nikolai, f35.4 | Germany | MT | Fresh | Yes | 96 | 91.2% (76.3%, 98.1%*) | 98.4% (91.3%*, 100%*) |
| [215] Nikolai, f35.2 | Germany | MT | Fresh | Yes | 132 | 86.1% (70.5%, 95.3%*) | 100% (96.2%*, 100%*) |
| [215] Nikolai, f35.1 | Germany | AN | Fresh | Yes | 132 | 86.1% (70.5%, 95.3%*) | 100% (96.2%*, 100%*) |
| [207] Lindner, f15.2 | Germany | MT | Fresh | Yes | 180 | 82.5% (67.2%*92.7%*) | 100% (96.5%, 100%) |
| [234] Stohr, f45.2 | Netherlands | AN | Fresh | Unclear | 1,611 | 61.5% (54.2%*, 68.4%*) | 99.7% (99.3%, 99.9%) |
| [271] Lindner, a53.2 | Germany | AN | Fresh | Yes | 179 | 80.5% (65.1%, 91.2%*) | 98.6% (94.9%, 99.8%*) |
| **Shenzhen Lvshiyuan Biotechnology, Green Spring SARS-CoV-2-Antigen-Schnelltest-Set (LFA)** | | | | | | | |
| [223] Pickering, f73.4 | UK | AN/OP | Banked | No | 200 | 77.0% (67.5%*, 84.8%*) | 98.0% (93.0%, 99.8%*) |
| **Shenzhen Bioeasy Biotechnology, 2019-nCov Antigen Rapid Test Kit (requires reader)** | | | | | | | |
| [285] Porte, a31.1 | Chile | NP/OP | Banked | No | 127 | 93.9% (86.3%*, 98.0%*) | 100% (92.1%*, 100%*) |
| [295] Weitzel, a41.4 | Chile | NP/OP | Banked | No | 111 | 85.0% (75.3%*, 92.0%*) | 100% (88.8%*, 100%) |
| [280] Parada-Ricart, a58.1 | Spain | NP | Fresh | Yes | 172 | 73.1%* (52.2%*, 88.4%*) | 85.6%* (78.9%*, 90.9%*) |
| [4] Krüger, a17.2 | Germany | NP/OP | Fresh | No | 727* | 66.7% (41.7%, 84.8%) | 93.1% (91.0%, 94.8%) |
| **Siemens Healthineers, CLINITEST Rapid COVID-19 Antigen Test (LFA)** | | | | | | | |
| [241] Torres, f29.1 | Spain | NP | Fresh | Yes | 178 | 80.2% (70.6%*, 87.8%*) | 100% (95.8%, 100%) |
| [241] Torres, f29.2 | Spain | NP | Fresh | Yes | 92 | 60.0% (38.7%*, 78.9%*) | 100% (94.6%, 100%) |
| [279] Olearo, a54.4 | Germany | OP | Unclear | No | 170 | 54.8%* (43.5%*, 65.7%*) | 100% (95.8%*, 100%) |
| [167] Baro, f33.2 | Spain | NP | Banked | No | 286 | 51.5% (41.3%, 61.6%) | 98.4% (95.3%, 99.7%*) |
| **Sugentech, SGTi-flex COVID-19 Ag (LFA)** | | | | | | | |
| [233] Shidlovskaya, f61.2 | Russia | NP | Fresh | Yes | 106 | 52.6% (40.9%, 64.0%) | 96.4% (81.7%, 99.9%) |
| **SureScreen Diagnostics, COVID-19 Rapid Antigen Visual Read (LFA)** | | | | | | | |
| [223] Pickering, f73.14 | UK | AN/OP | Banked | No | 23 | 74.0%* (51.6%*, 89.8%*) | Not provided |
| [223] Pickering, f73.3 | UK | AN/OP | Banked | No | 200 | 65.0% (54.8%*, 74.3%*) | 100% (96.4%*, 100%*) |
| [223] Pickering, f73.15 | UK | AN/OP | Banked | No | 23 | 65.2% (42.7%*, 83.6%*) | Not provided |
| [223] Pickering, f73.13 | UK | AN/OP | Banked | No | 23 | 61.0% (38.5%*, 80.3%*) | Not provided |
| [167] Baro, f33.5 | Spain | NP | Banked | No | 286 | 28.8% (20.2%, 38.6%) | 97.8% (94.5%, 99.4%) |
| **SureScreen Diagnostics, COVID-19 Rapid Antigen Fluorescent (requires reader)** | | | | | | | |
| [223] Pickering, f73.6 | UK | AN/OP | Banked | No | 200 | 69.0% (59.0%*, 77.9%*) | 98.0% (93%, 99.8%*) |
| [223] Pickering, f73.7 | UK | AN/OP | Banked | No | 141 | 60.3% (51.7%*, 68.4%*) | Not provided |
| **VivaCheck, VivaDiag SARS-CoV-2 Ag Rapid Test (LFA)** | | | | | | | |
| [194] Homza, f87.5 | Czech Republic | NP | Fresh | Yes | 268 | 41.8% (31.5%, 52.6%) | 96.0% (92.0%, 98.4%) |

*(Continued)*

**Table 1.** (*Continued*)

| Reference, first author, dataset ID | Study location | Sample type | Sample condition | IFU-conforming | Sample size | Sensitivity (95% CI) | Specificity (95% CI) |
|---|---|---|---|---|---|---|---|
| **Zhuhai Encode Medical Engineering, SARS-CoV-2 Antigen Rapid Test (LFA)** | | | | | | | |
| [223] Pickering, f73.5 | UK | AN/OP | Banked | No | 200 | 74.0% (64.3%*, 82.3%*) | 100% (96.4%*, 100%) |
| [223] Pickering, f73.9 | UK | AN/OP | Banked | No | 90 | 74.4% (64.2%*, 83.1%*) | Not provided |

Datasets with an underlined reference and first author had not undergone peer-review yet at the time of data extraction (1 May 2021). In datasets with an underlined sample size, the samples were used in head-to-head studies, i.e., performing different Ag-RDTs on the same patient.

*Values differ from those provided in the respective paper due to missing or contradictory data. A list including the original data can be found in S2 Table.

AN, anterior nasal; BAL/TW, bronchoalveolar lavage and throat wash; CI, confidence interval; IFU, instructions for use; FIND, Foundation for Innovative New Diagnostics; LFA, lateral flow assay; MT, mid-turbinate; NP, nasopharyngeal; OP, oropharyngeal.

CI 60.6% to 70.8%). Pooled specificity was similar in both groups (99.1% [95% CI 98.8–99.4%] and 98.3% [95% CI 97.7% to 98.8%], respectively).

## Analysis of specific tests

Based on 119 datasets with 71,424 tests performed, we were able to perform bivariate meta-analysis of the sensitivity and specificity for 12 different Ag-RDTs (Fig 4). Across these, the pooled estimates of sensitivity and specificity on all samples were 72.1% (95% CI 68.8% to

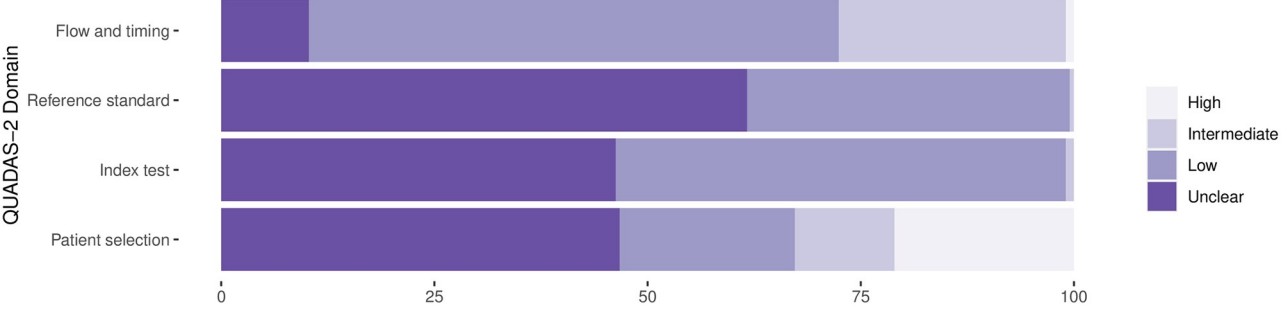

**Fig 2. Methodological quality of the clinical accuracy studies: Risk of bias.** Proportion of studies with low, intermediate, high, or unclear risk of bias (percent).

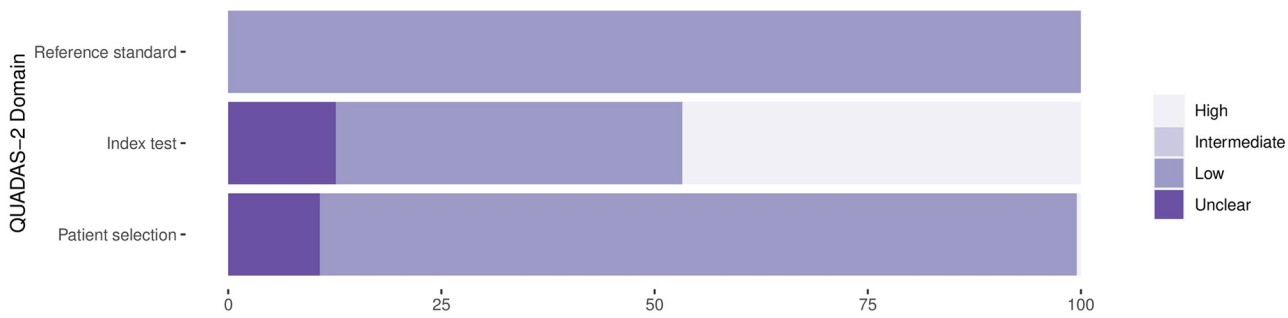

**Fig 3. Methodological quality of the clinical accuracy studies: Applicability.** Proportion of studies with low, intermediate, high, or unclear concerns regarding applicability (percent).

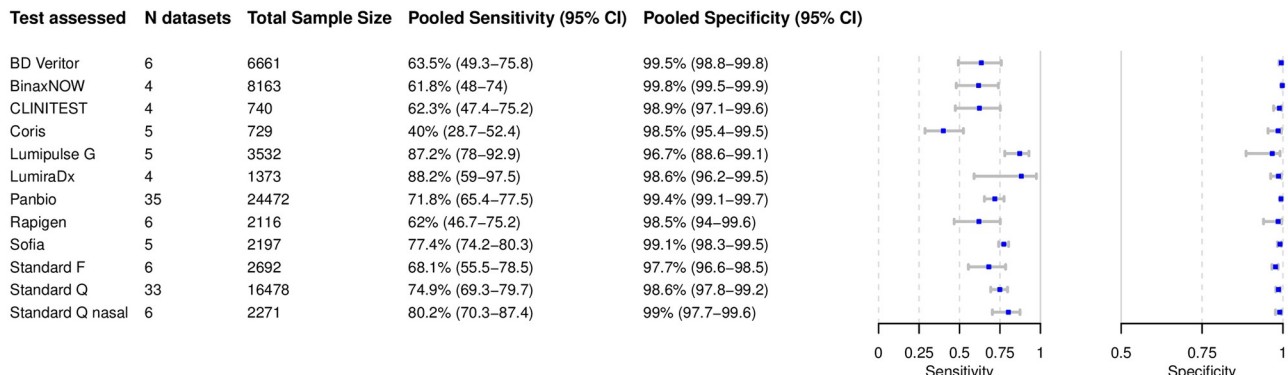

| Test assessed | N datasets | Total Sample Size | Pooled Sensitivity (95% CI) | Pooled Specificity (95% CI) |
|---|---|---|---|---|
| BD Veritor | 6 | 6661 | 63.5% (49.3–75.8) | 99.5% (98.8–99.8) |
| BinaxNOW | 4 | 8163 | 61.8% (48–74) | 99.8% (99.5–99.9) |
| CLINITEST | 4 | 740 | 62.3% (47.4–75.2) | 98.9% (97.1–99.6) |
| Coris | 5 | 729 | 40% (28.7–52.4) | 98.5% (95.4–99.5) |
| Lumipulse G | 5 | 3532 | 87.2% (78–92.9) | 96.7% (88.6–99.1) |
| LumiraDx | 4 | 1373 | 88.2% (59–97.5) | 98.6% (96.2–99.5) |
| Panbio | 35 | 24472 | 71.8% (65.4–77.5) | 99.4% (99.1–99.7) |
| Rapigen | 6 | 2116 | 62% (46.7–75.2) | 98.5% (94–99.6) |
| Sofia | 5 | 2197 | 77.4% (74.2–80.3) | 99.1% (98.3–99.5) |
| Standard F | 6 | 2692 | 68.1% (55.5–78.5) | 97.7% (96.6–98.5) |
| Standard Q | 33 | 16478 | 74.9% (69.3–79.7) | 98.6% (97.8–99.2) |
| Standard Q nasal | 6 | 2271 | 80.2% (70.3–87.4) | 99% (97.7–99.6) |

**Fig 4. Bivariate analysis of 12 antigen rapid diagnostic tests.** Pooled sensitivity and specificity were calculated based on reported sample sizes, true positives, true negatives, false positives, and false negatives.

75.3%) and 99.0% (95% CI 98.7% to 99.2%), respectively, which were very similar to the overall pooled estimates across all meta-analyzed datasets (71.2% and 98.9%, respectively, above).

The highest pooled sensitivity was found for the SARS-CoV-2 Antigen Test by LumiraDx (UK; henceforth called LumiraDx) and the Lumipulse G SARS-CoV-2 Ag by Fujirebio (Japan; henceforth called Lumipulse G), with 88.2% (95% CI 59.0% to 97.5%) and 87.2% (95% CI 78.0% to 92.9%), respectively. The Sofia SARS Antigen FIA by Quidel (California, US; henceforth called Sofia) had a pooled sensitivity of 77.4% (95% CI 74.2% to 80.3%). Of the non-instrument tests, the Standard Q and the Standard Q nasal test by SD Biosensor (South Korea; distributed in Europe by Roche, Germany; henceforth called Standard Q nasal) performed best, with a pooled sensitivity of 74.9% (95% CI 69.3% to 79.7%) and 80.2% (95% CI 70.3% to 87.4%), respectively. The pooled sensitivity for Panbio was 71.8% (95% CI 65.4% to 77.5%). Of all Ag-RDTs, the COVID-19 Ag Respi-Strip by Coris BioConcept (Belgium; henceforth called Coris) had the lowest pooled sensitivity, 40.0% (95% CI 28.7% to 52.4%).

The pooled specificity was above 98% for all of the tests, except for the Standard F by SD Biosensor (South Korea) and Lumipulse G, with specificities of 97.7% (95% CI 96.6% to 98.5%) and 96.7% (95% CI 88.6% to 99.1%), respectively. Hierarchical summary receiver operating characteristic values for Standard Q and LumiraDx are available in S2 Fig.

Three Ag-RDTs did not have sufficient data to allow for a bivariate meta-analysis, so a univariate analysis was conducted (Fig 5). For the INNOVA SARS-CoV-2 Antigen Rapid Qualitative Test by Innova Medical Group (California, US), this resulted in a pooled sensitivity and specificity of 76.1% (95% CI 68.1% to 84.1%) and 99.4% (95% CI 98.7% to 100%), respectively. For the NADAL by nal von minden (Germany) and the COVID-19 Rapid Antigen Visual Read by SureScreen Diagnostics (UK), sufficient data were available to analyze only sensitivity, resulting in pooled sensitivity estimates of 58.4% (95% CI 29.2% to 87.6%) and 58.0% (95% CI 38.3% to 77.6%), respectively.

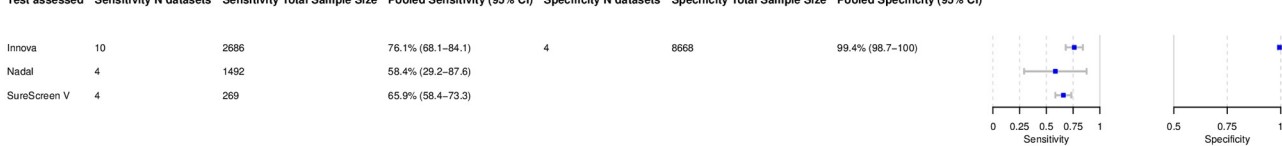

| Test assessed | Sensitivity N datasets | Sensitivity Total Sample Size | Pooled Sensitivity (95% CI) | Specificity N datasets | Specificity Total Sample Size | Pooled Specificity (95% CI) |
|---|---|---|---|---|---|---|
| Innova | 10 | 2686 | 76.1% (68.1–84.1) | 4 | 8668 | 99.4% (98.7–100) |
| Nadal | 4 | 1492 | 58.4% (29.2–87.6) | | | |
| SureScreen V | 4 | 269 | 65.9% (58.4–73.3) | | | |

**Fig 5. Univariate analysis of 3 antigen rapid diagnostic tests.** Pooled sensitivity and specificity were calculated based on reported sensitivity, specificity, and confidence intervals. SureScreen V, SureScreen Diagnostics COVID-19 Rapid Antigen Visual Read.

**Table 2. Summary clinical accuracy data for major Ag-RDTs not included in the meta-analysis.**

| Manufacturer, Ag-RDT | Number of datasets | Sensitivity range | Specificity range | Comments |
|---|---|---|---|---|
| Bionote, NowCheck (LFA) | 3 | 55.6% to 89.9% | 97.3% to 100% | • Two of the studies were IFU-conforming, whereas IFU conformity for the study reporting 55.6% sensitivity was unclear |
| Denka, Quick Navi (LFA) | 2 | 72.5% to 86.7% | 100%* | • Both studies were conducted on fresh samples, but for the one reporting 72.5% IFU conformity was unclear |
| Fujirebio, ESPLINE SARS-CoV-2 (LFA) | 3 | 8.1% to 80.7% | 100%* | • The dataset reporting 8.1% sensitivity used saliva samples (not IFU-conforming) and the majority of samples showed a Ct value > 25 |
| JOYSBIO Biotechnology, COVID-19 Antigen Rapid Test Kit (LFA) | 2 | 57.8% to 70.5% | 98.5% to 99.1% | • The datasets used NP and AN samples, respectively; both were performed by IFU on symptomatic people or high-risk contacts |
| MEDsan, SARS-CoV-2 Antigen Rapid Test (LFA) | 2 | 36.5% to 45.2% | 97% to 99.6% | • Both studies were conducted on OP samples, which is IFU-conforming for this test |
| R-Biopharm, RIDA QUICK SARS-CoV-2 Antigen (LFA) | 3 | 39.2% to 77.6% | 96.2% to 100% | • Two datasets originate from the same study and no study was conducted as per IFU<br>• The dataset reporting 39.2% included only asymptomatic persons with Ct values between 22.1 and 36.4 |
| Shenzhen Bioeasy Biotechnology, 2019-nCov Antigen Rapid Test Kit (requires reader) | 4 | 66.7% to 93.9% | 85.6% to 100% | • The dataset reporting 85.6% specificity was IFU-conforming<br>• The datasets reporting highest sensitivity were drawn from just symptomatic patients; for the others, symptomatic patients made up more than two-thirds of the study population |
| SureScreen Diagnostics, COVID-19 Rapid Antigen Fluorescent (requires reader) | 2 | 60.3% to 69.0% | 98%* | • Both datasets originate from the same study and were not IFU-conforming, conducted on stored samples |
| Zhuhai Encode Medical Engineering, SARS-CoV-2 Antigen Rapid Test (LFA) | 2 | 74.0% to 74.4% | 100%* | • Both datasets originate from the same study, a retrospective head-to-head comparison<br>• Stored AN/MT samples were assessed |

*Only 1 dataset for specificity was provided.

Ag-RDT, antigen rapid diagnostic test; AN, anterior nasal; Ct, cycle threshold; IFU, instructions for use; LFA, lateral flow assay; MT, mid-turbinate; NP, nasopharyngeal; OP, oropharyngeal.

The remaining 35 Ag-RDTs did not present sufficient data for univariate or bivariate meta-analysis. However, 9/35 had results presented in more than 1 dataset, and these are summarized in Table 2. Herein, the widest ranges of sensitivity were found for the ESPLINE SARS-CoV-2 by Fujirebio (Japan), with sensitivity reported between 8.1% and 80.7%, and the RIDA QUICK SARS-CoV-2 Antigen by R-Biopharm (Germany), with sensitivity between 39.2% and 77.6%, both with 3 datasets each. In contrast, 2 other tests with 2 datasets each showed the least variability in sensitivity: The Zhuhai Encode Medical Engineering SARS-CoV-2 Antigen Rapid Test (China) reported sensitivity between 74.0% and 74.4%, and the COVID-19 Rapid Antigen Fluorescent by SureScreen Diagnostics (UK) reported sensitivity between 60.3% and 69.0%. However, for both tests, both datasets originated from the same studies. Overall, the lowest sensitivity range was reported for the SARS-CoV-2 Antigen Rapid Test by MEDsan (Germany): 36.5% to 45.2% across 2 datasets. The specificity ranges were above 96% for most of the tests. A notable outlier was the 2019-nCov Antigen Rapid Test Kit by Shenzhen Bioeasy Biotechnology (China; henceforth called Bioeasy), reporting the worst, with a specificity as low as 85.6% in 1 study. Forest plots for the datasets for each Ag-RDT are provided in S3 Fig. The remaining 26 Ag-RDTs that were evaluated in 1 dataset only are included in Table 1 S3 Fig.

In total, 16 studies, accounting for 53 datasets, conducted head-to-head clinical accuracy evaluations of different tests using the same samples from the same participants. These datasets have underlined sample sizes in Table 1; 15 such studies included more than 100 samples, and

1 study included too few samples to draw clear conclusions [286]. Four studies performed their head-to-head evaluation as per manufacturers' instructions and on symptomatic patients. Across 3 of them, Standard Q (sensitivity 73.2% to 91.2%) and Standard Q nasal (sensitivity 82.5% to 91.2%) showed a similar range of sensitivity [207,215,271]. The fourth reported a sensitivity of 56.4% (95% CI 44.7% to 67.6%) for the Biocredit Covid-19 Ag by RapiGEN (South Korea; henceforth called Rapigen) and 52.6% (95% CI 40.9% to 64.0%) for the SGTi-flex COVID-19 Ag by Sugentech (South Korea) [233].

All other head-to-head comparisons were not IFU-conforming. In one of these, the Rapid COVID-19 Ag Test by Healgen (sensitivity 77.1%) performed better than Standard Q and Panbio (sensitivity 69.8% and 67.7%, respectively) [178]. In contrast to the overall findings of the meta-analysis above, 2 other head-to-head studies found that both Standard Q (sensitivity 43.6% and 49.4%) and Panbio (sensitivity 38.6% and 44.6%) had lower performance than the CLINITEST Rapid COVID-19 Antigen Test by Siemens Healthineers (Germany; henceforth called Clinitest), with reported sensitivity of 51.5% and 54.9% [167,279]. However, another study found both Standard Q and Panbio (sensitivity 81.0% and 82.9%, respectively) to have a higher accuracy than Sofia (sensitivity 80.4%) [196].

## Subgroup analyses

The results are presented in Figs 6–10. Detailed results for the subgroup analyses are available in S4–S9 Figs.

**Subgroup analysis by Ct values.** High sensitivity was achieved for Ct value < 20, at 96.5% (95% CI 92.6% to 98.4%). The pooled sensitivity for Ct value < 25 was markedly better, at 95.8% (95% CI 92.3% to 97.8%), compared to the group with Ct value ≥ 25, at 50.7% (95% CI 35.6% to 65.8%). A similar pattern was observed when the Ct values were analyzed using the cutoffs <30 and ≥30, resulting in a sensitivity of 79.9% (95% CI 70.3% to 86.9%) and 20.9% (95% CI 12.5% to 32.8%), respectively (Fig 6).

| Ct Values | N datasets | Total Sample Size | Pooled Sensitivity (95% CI) |
|---|---|---|---|
| <20 | 22 | 741 | 96.5% (92.6–98.4) |
| >=20 | 12 | 598 | 94.4% (79.2–98.7) |
| <25 | 47 | 3004 | 95.8% (92.3–97.8) |
| >=25 | 24 | 987 | 50.7% (35.6–65.8) |
| <30 | 37 | 2879 | 79.9% (70.3–86.9) |
| >=30 | 29 | 509 | 20.9% (12.5–32.8) |

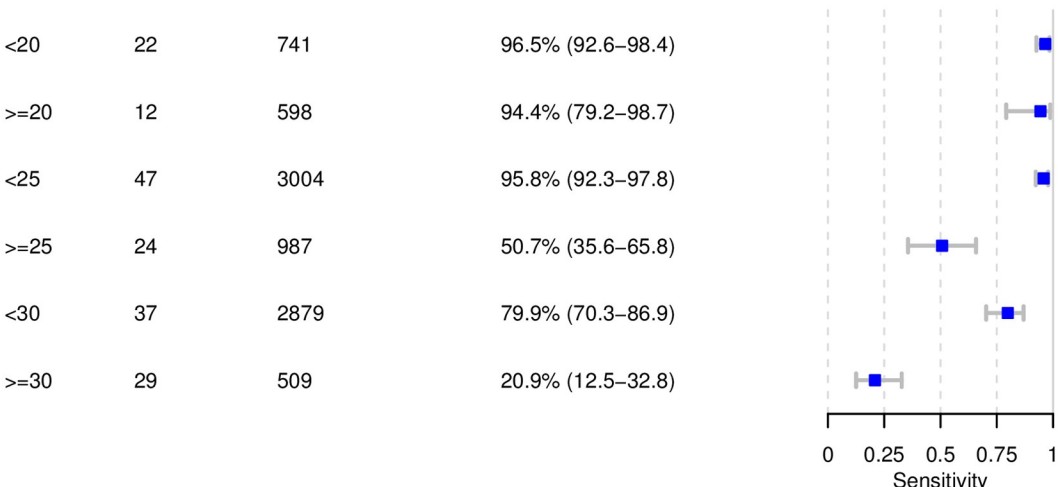

**Fig 6. Pooled sensitivity by cycle threshold (Ct) values.** Low Ct values are the reverse transcription PCR semi-quantitative correlate for a high virus concentration.

| Testing procedure | N datasets | Total Sample Size | Pooled Sensitivity (95% CI) | Pooled Specificity (95% CI) |
|---|---|---|---|---|
| IFU | 81 | 49643 | 76.3% (73.1–79.2) | 99.1% (98.8–99.4) |
| non–IFU | 75 | 31416 | 65.9% (60.6–70.8) | 98.3% (97.7–98.8) |
| unclear | 17 | 14288 | 67.1% (55.8–76.7) | 99.4% (99.2–99.6) |

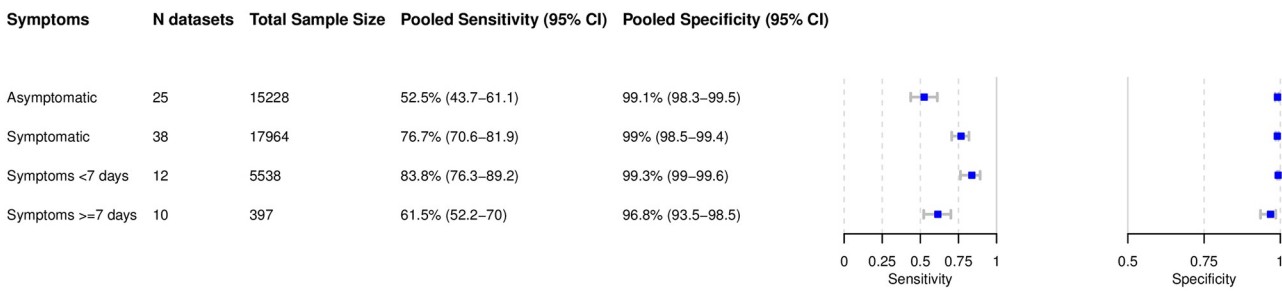

**Fig 7. Pooled sensitivity and specificity by instructions for use (IFU) conformity.**

| Sample type | N datasets | Total Sample Size | Pooled Sensitivity (95% CI) | Pooled Specificity (95% CI) |
|---|---|---|---|---|
| AN/MT | 32 | 25814 | 75.5% (70.4–79.9) | 99.2% (98.7–99.5) |
| NP | 122 | 59810 | 71.6% (68.1–74.9) | 98.9% (98.5–99.1) |
| OP | 7 | 5165 | 53.1% (40.9–65) | 99.3% (98.2–99.7) |
| saliva | 4 | 1088 | 37.9% (11.8–73.5) | 99.2% (97.5–99.7) |
| unclear | 5 | 3251 | 77.4% (71.8–82.2) | 99.1% (97–99.7) |

**Fig 8. Pooled sensitivity and specificity by sample type.** AN, anterior nasal; MT, mid-turbinate; NP, nasopharyngeal; OP, oropharyngeal.

| Symptoms | N datasets | Total Sample Size | Pooled Sensitivity (95% CI) | Pooled Specificity (95% CI) |
|---|---|---|---|---|
| Asymptomatic | 25 | 15228 | 52.5% (43.7–61.1) | 99.1% (98.3–99.5) |
| Symptomatic | 38 | 17964 | 76.7% (70.6–81.9) | 99% (98.5–99.4) |
| Symptoms <7 days | 12 | 5538 | 83.8% (76.3–89.2) | 99.3% (99–99.6) |
| Symptoms >=7 days | 10 | 397 | 61.5% (52.2–70) | 96.8% (93.5–98.5) |

**Fig 9. Pooled sensitivity and specificity by presence of symptoms and symptom duration.**

In addition, it was possible to meta-analyze test-specific pooled sensitivity for Panbio: 97.7% sensitivity (95% CI 95.3% to 98.9%) for Ct value < 20, 95.8% (95% CI 92.3% to 97.8%) for Ct value < 25, and 83.4% (95% CI 69.1% to 91.9%) for Ct value < 30. Sensitivity was 61.2% (95% CI 38.8% to 79.7%) for Ct value ≥ 25 and 30.5% (95% CI 16.0% to 50.4%) for Ct value ≥ 30. For the other Ag-RDTs only limited data were available, which are presented in S5 Fig.

**Subgroup analysis by IFU conformity.** The summary results are presented in Fig 7. When assessing only studies with IFU-conforming testing, pooled sensitivity from 81 datasets with 49,643 samples was 76.3% (95% CI 73.1% to 79.2%). When non-IFU-conforming sampling (75 datasets, 31,416 samples) was performed, sensitivity decreased to 65.9% (95% CI 60.6% to 70.8%).

| | N datasets | Total Sample Size | Pooled Sensitivity (95% CI) | Pooled Specificity (95% CI) |
|---|---|---|---|---|
| Age <18 years | 7 | 3837 | 64.3% (54.7–72.9) | 99.4% (98.9–99.7) |
| Age >=18 years | 16 | 7326 | 74.8% (66.5–81.6) | 98.7% (97.2–99.4) |

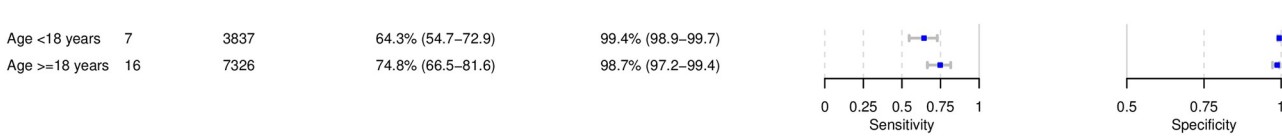

**Fig 10. Pooled sensitivity and specificity by age.**

For 5 tests it was possible to calculate pooled sensitivity estimates including only datasets with IFU-conforming testing: Panbio (sensitivity 76.5% [95% CI 69.5% to 82.3%]; 17 datasets, 12,856 samples), Standard Q (sensitivity 79.3% [95% CI 73.5% to 84.1%]; 15 datasets, 6,584 samples), BinaxNOW (sensitivity 61.8% [95% CI 48.0% to 74.0%]; 4 datasets, 8,163 samples), Rapigen (sensitivity 67.1% [95% CI 50.4% to 80.4%]; 4 datasets, 1,934 samples), and Standard Q nasal (sensitivity 83.8% [95% CI 77.8% to 88.4%]; 5 datasets, 683 samples). Specificity was above 98.6% for all tests.

In contrast, when the Panbio (14 datasets, 9,233 samples) and Standard Q (14 datasets, 4,714 samples) tests were not performed according to IFU, pooled sensitivity decreased to 64.3% (95% CI 50.9% to 75.8%) and 67.4% (95% CI 57.2% to 76.2%), respectively.

**Subgroup analysis by sample type.**   Most datasets evaluated NP or combined NP/OP swabs (122 datasets and 59,810 samples) as the sample type for the Ag-RDT. NP or combined NP/OP swabs achieved a pooled sensitivity of 71.6% (95% CI 68.1% to 74.9%). Datasets that used AN/MT swabs for Ag-RDTs (32 datasets and 25,814 samples) showed a summary estimate for sensitivity of 75.5% (95% CI 70.4% to 79.9%). This was confirmed by 2 studies that reported direct head-to-head comparison of NP and MT samples from the same participants using the same Ag-RDT (Standard Q), where the 2 sample types showed equivalent performance [271,272]. Analysis of performance with an OP swab (7 datasets, 5,165 samples) showed a pooled sensitivity of only 53.1% (95% CI 40.9% to 65.0%). Saliva swabs (4 datasets, 1,088 samples) showed the lowest pooled sensitivity, at only 37.9% (95% CI 11.8% to 73.5%) (Fig 8).

We were not able to perform a subgroup meta-analysis for BAL/TW due to insufficient data: There was only 1 study with 73 samples evaluating Rapigen, Panbio, and Standard Q [286]. However, BAL/TW would in any case be considered an off-label use.

**Subgroup analysis in symptomatic and asymptomatic patients.**   Within the datasets possible to meta-analyze, 17,964 (54.1%) samples were from symptomatic, and 15,228 (45.9%) from asymptomatic, patients. The pooled sensitivity for symptomatic patients was markedly different from that of asymptomatic patients: 76.7% (95% CI 70.6% to 81.9%) versus 52.5% (95% CI 43.7% to 61.1%). Specificity was 99% for both groups (Fig 9). Median Ct values differed in symptomatic and asymptomatic patients. For those studies where it was possible to extract a median Ct value, it ranged from 20.5 to 27.0 in symptomatic patients [170,207,226,258,271,272] and from 27.2 to 30.5 in asymptomatic patients [170,201,258].

**Subgroup analysis comparing symptom duration.**   Data were analyzed for 5,538 patients with symptoms less than 7 days, but very limited data were available for patients with symptoms ≥7 days (397 patients). The pooled sensitivity for patients with onset of symptoms <7 days was 83.8% (95% CI 76.3% to 89.2%), which is markedly higher than the 61.5% (95% CI 52.2% to 70.0%) sensitivity for individuals tested ≥7 days from onset of symptoms (Fig 9).

**Subgroup analysis by age.**   For adult patients (age ≥ 18 years), it was possible to pool estimates across 3,837 samples, whereas the pediatric group (age < 18 years) included 7,326 samples. Sensitivity and specificity were 64.3% (95% CI 54.7% to 72.9%) and 99.4% (95% CI 98.9% to 99.7%), respectively, in mostly symptomatic patients aged <18 years. In patients aged ≥18 years, sensitivity increased to 74.8% (95% CI 66.5% to 81.6%), while the specificity was similar (98.7%, 95% CI 97.2% to 99.4%) (Fig 10).

**Subgroup analysis by type of RT-PCR and viral load.**   We were not able to perform a meta-analysis for the subgroups by type of RT-PCR or viral load (viral copies/mL) due to insufficient data.

In 152 (71.0%) of the datasets only 1 type of RT-PCR was used, whereas 37 (17.3%) of the datasets tested samples in the same dataset using different RT-PCR methods. For 25 (11.7%) of the datasets, the type of RT-PCR was not reported. The Cobas SARS-CoV-2 Test from Roche (Germany) was used most frequently, in 63 (29.4%) of the datasets, followed by the Allplex

2019-nCoV Assay from Seegene in 41 (19.2%) and the SARS-CoV-2 assay from TaqPath in 20 (9.3%) of the datasets.

Median sensitivity was 72.4% (range 46.9% to 100%) in samples with viral load > 5 log10 copies/mL, 97.8% (range 71.4% to 100%) for >6 log10 copies/mL, and 100% (range 93.8% to 100%) for >7 log10 copies/mL, showing that the sensitivity increases with increasing viral load.

**Meta regression.** We were not able to perform a meta-regression due to the considerable heterogeneity in reporting subgroups, which resulted in too few studies with sufficient data for comparison.

**Publication bias.** The result of the Deeks test ($p = 0.001$) shows significant asymmetry in the funnel plot for all datasets with complete results. This indicates there may be publication bias from studies with small sample sizes. The funnel plot is presented in S10 Fig.

### Comparison with analytical studies

The 9 analytical studies were divided into 63 datasets, evaluating 23 different Ag-RDTs. Only 7 studies reported a sample size, for which 833 (90.6%) samples originated from NP swabs, while for 86 (9.4%) the sample type was unclear. One of the 2 studies not reporting sample size used saliva samples [198], while the other used the sample type specified in the respective Ag-RDT's IFU [173].

Overall, the reported analytical sensitivity (limit of detection [LOD]) in the studies resembled the results of the meta-analysis presented above. Rapigen (LOD, in log10 copies per swab: 10.2) and Coris (LOD 7.46) were found to perform worse than Panbio (LOD 6.6 to 6.1) and Standard Q (LOD 6.8 to 6.0), whereas Clinitest (LOD 6.0) and BinaxNOW by Abbott (LOD 4.6 to 4.9) performed better [191,256,282]. Similar results were found in another study, where Standard Q showed the lowest LOD (detecting virus up to what is an equivalent Ct value of 26.3 to 28.7), compared to that of Rapigen and Coris (detecting virus up to what is an equivalent Ct value of only 18.4 for both) [208,274,275]. However, another study found Panbio, Standard Q, Coris, and BinaxNOW to have a similar LOD values of $5.0 \times 10^3$ plaque forming units (PFU)/mL, but the ESPLINE SARS-CoV-2 by Fujirebio (Japan), the COVID-19 Rapid Antigen Test by Mologic (UK), and the Sure Status COVID-19 Antigen Card Test by Premier Medical Corporation (India) performed markedly better (LOD $2.5 \times 10^2$ to $5.0 \times 10^2$ PFU/mL) [173]. An overview of all LOD values reported in the studies can be found in S3 Table.

### Sensitivity analysis

When the datasets from case–control studies (25/173) were excluded, the estimated sensitivity did not differ greatly, with a value of 70.9% (95% CI 67.7% to 73.9%), compared to 71.2% (95% CI 68.2% to 74.0%) in the overall analysis, with no change in pooled specificity. When the datasets from preprints (64/173) were excluded, sensitivity decreased slightly, to 67.2% (95% CI 62.9% to 71.3%), compared to the overall analysis.

### Discussion

In this comprehensive systematic review and meta-analysis, we have summarized the data of 133 studies evaluating the accuracy of 61 different Ag-RDTs. Across all meta-analyzed samples, our results show a pooled sensitivity and specificity of 71.2% (95% CI 68.2% to 74.0%) and 98.9% (95% CI 98.6% to 99.1%), respectively. Over half of the studies did not perform the Ag-RDT in accordance with the test manufacturers' recommendation, or the performance was unknown, which negatively impacted the sensitivity. When we considered only

IFU-conforming studies, the sensitivity increased to 76.3% (95% CI 73.1% to 79.2%). While we found the sensitivity to vary across specific tests, the specificity was consistently high.

The 2 Ag-RDTs that have been approved through the WHO emergency use listing procedure, Abbott Panbio and SD Biosensor Standard Q (distributed by Roche in Europe), have not only drawn the largest research interest, but also perform at or above average when their pooled accuracy is compared to that of all Ag-RDTs (sensitivity of 71.8% for Panbio and 74.9% for Standard Q). Standard Q nasal demonstrated an even higher pooled sensitivity (80.2% compared to the NP test), although this is likely due to variability in the populations tested, as head-to-head performance showed a comparable sensitivity. Three other Ag-RDTs showed an even higher accuracy, with sensitivities ranging from 77.4% to 88.2% (namely Sofia, Lumipulse G, and LumiraDx), but were only assessed on relatively small samples sizes (ranging from 1,373 to 3,532), and all required an instrument/reader.

Not surprisingly, lower Ct values, the RT-PCR semi-quantitative correlate for high virus concentration, resulted in significantly higher Ag-RDT sensitivity than higher Ct values (pooled sensitivity 96.5% and 95.8% for Ct value < 20 and <25, respectively, versus 50.7% and 20.9% for Ct value ≥ 25 and ≥30, respectively). This confirms prior data that suggested that antigen concentrations and Ct values were highly correlated in NP samples [16]. Ag-RDTs also showed higher sensitivity in patients within 7 days after symptom onset compared to patients later in the course of the disease (pooled sensitivity 83.8% versus 61.5%), which is to be expected given that samples from patients within the first week after symptom onset have been shown to contain the highest virus concentrations [298]. In line with this, studies reporting an unexpectedly low overall sensitivity either shared a small population size with an on average high Ct value [230,273,288] or performed the Ag-RDT not as per IFU, e.g., using saliva or prediluted samples [167,170,203,248,279]. In contrast, studies with an unusually high Ag-RDT sensitivity were based on study populations with a low median Ct value, between 18 and 22 [189,255,284].

Our analysis also found that the accuracy of Ag-RDTs is substantially higher in symptomatic patients than in asymptomatic patients (pooled sensitivity 76.7% versus 52.5%). This is not surprising as studies that enrolled symptomatic patients showed a lower range of median Ct values (i.e., higher viral load) than studies enrolling asymptomatic patients. Given that other studies found symptomatic and asymptomatic patients to have comparable viral loads [299,300], the differences found in our analysis are likely explained by the varied time in the course of the disease at which testing is performed in asymptomatic patients presenting for one-time screening testing. Because symptoms start in the early phase of the disease, when viral load is still high, studies testing only symptomatic patients have a higher chance of including patients with high viral loads. In contrast, study populations drawn from only asymptomatic patients have a higher chance of including patients at any point of disease (i.e., including late in disease, when PCR is still positive, but viable virus is rapidly decreasing) [301].

With regards to the sampling and testing procedure, we found Ag-RDTs to perform similarly across upper respiratory swab samples (e.g., NP and AN/MT), particularly when considering the most reliable comparisons from head-to-head studies.

Similar to previous assessment [7], the methodological quality of the included studies revealed a very heterogenous picture. In the future, aligning the design of clinical accuracy studies with common agreed-upon minimal specifications (e.g., by WHO or the European Centre for Disease Control and Prevention) and reporting the results in a standardized way [302] would improve data quality and comparability.

The main strengths of our study lie in its comprehensive approach and continuous updates. By linking this review to our website, https://www.diagnosticsglobalhealth.org, we strive to equip decision makers with the latest research findings on Ag-RDTs for SARS-CoV-2 and, to

the best of our knowledge, are the first in doing so. At least once per week the website is updated by continuing the literature search and process described above. We plan to update the meta-analysis on a monthly basis and publish it on the website. Furthermore, our study used rigorous methods as both the study selection and data extraction were performed by one author and independently validated by a second, we conducted blinded pilot extractions before of the actual data extraction, and we prepared a detailed interpretation guide for the QUADAS-2 tool.

The study may be limited by the inclusion of both preprints and peer-reviewed literature, which could affect the quality of the data extracted. However, we aimed to balance this potential effect by applying a thorough assessment of all clinical studies included, utilizing the QUADAS-2 tool, and performing a sensitivity analysis excluding preprint manuscripts. In addition, the studies included in our analysis varied widely in the reported range of viral loads, limiting the comparability of their results. To control for this, we analyzed the Ag-RDTs' performance at different levels of viral load. Finally, even though we are aware that further data exist from other sources, for example from governmental research institutes [303], such data could not be included because sufficiently detailed descriptions of the methods and results are not publicly available.

## Conclusion

In summary, it can be concluded that there are Ag-RDTs available that have high sensitivity for the detection of a SARS-CoV-2 infection—particularly when performed in the first week of illness, when viral load is high—and excellent specificity. However, our analysis also highlights the variability in results between tests (which is not reflected in the manufacturer-reported data), indicating the need for independent validations. Furthermore, the analysis highlights the importance of performing tests in accordance with the manufacturers' recommended procedures, and in alignment with standard diagnostic evaluation and reporting guidelines. The accuracy achievable by the best-performing Ag-RDTs, combined with the rapid turnaround time compared to RT-PCR, suggests that these tests could have a significant impact on the pandemic if applied in thoughtful testing and screening strategies.

## Supporting information

**S1 Fig. Detailed results of the QUADAS-2 assessment.**
(PDF)

**S2 Fig. Hierarchical summary receiver operating characteristic curve for Standard Q Ag-RDT.**
(PDF)

**S3 Fig. Forest plots of all Ag-RDTs.**
(PDF)

**S4 Fig. Forest plots for subgroup analysis by Ct value.**
(PDF)

**S5 Fig. Forest plots for subgroup analysis by Ct value per test.**
(PDF)

**S6 Fig. Forest plots for subgroup analysis by IFU versus non-IFU.**
(PDF)

**S7 Fig. Forest plots for subgroup analysis by sample type.**
(PDF)

**S8 Fig. Forest plots for subgroup analysis by symptomatic versus asymptomatic.**
(PDF)

**S9 Fig. Forest plots for subgroup analysis by symptom duration.**
(PDF)

**S10 Fig. Funnel plot test for all datasets included in the meta-analysis.**
(PDF)

**S1 PRISMA Checklist.**
(DOCX)

**S1 Table. List of data items extracted from studies.**
(XLSX)

**S2 Table. List of original data.**
(XLSX)

**S3 Table. Summary of analytical studies.**
(XLSX)

**S1 Text. Study protocol submitted to PROSPERO (registration: CRD42020225140).**
(DOCX)

**S2 Text. Search strategy.**
(DOCX)

**S3 Text. QUADAS-2 assessment interpretation guide.**
(DOCX)

## Author Contributions

**Conceptualization:** Lukas E. Brümmer, Stephan Katzenschlager, Mary Gaeddert, Claudia M. Denkinger.

**Data curation:** Lukas E. Brümmer, Stephan Katzenschlager, Mary Gaeddert, Christian Erdmann, Stephani Schmitz, Marc Bota, Aurélien Macé, Claudia M. Denkinger.

**Formal analysis:** Lukas E. Brümmer, Stephan Katzenschlager, Mary Gaeddert, Christian Erdmann, Stephani Schmitz, Marc Bota, Claudia M. Denkinger.

**Funding acquisition:** Claudia M. Denkinger.

**Investigation:** Lukas E. Brümmer, Stephan Katzenschlager, Christian Erdmann, Stephani Schmitz, Marc Bota, Maurizio Grilli, Claudia M. Denkinger.

**Methodology:** Lukas E. Brümmer, Stephan Katzenschlager, Mary Gaeddert, Nira R. Pollock, Aurélien Macé, Sergio Carmona, Stefano Ongarello, Jilian A. Sacks, Claudia M. Denkinger.

**Project administration:** Lukas E. Brümmer, Claudia M. Denkinger.

**Resources:** Lukas E. Brümmer, Claudia M. Denkinger.

**Software:** Mary Gaeddert, Maurizio Grilli.

**Supervision:** Lukas E. Brümmer, Claudia M. Denkinger.

**Validation:** Lukas E. Brümmer, Mary Gaeddert, Aurélien Macé, Claudia M. Denkinger.

**Visualization:** Mary Gaeddert.

**Writing – original draft:** Lukas E. Brümmer, Stephan Katzenschlager, Mary Gaeddert, Claudia M. Denkinger.

**Writing – review & editing:** Lukas E. Brümmer, Stephan Katzenschlager, Mary Gaeddert, Christian Erdmann, Stephani Schmitz, Marc Bota, Maurizio Grilli, Jan Larmann, Markus A. Weigand, Nira R. Pollock, Aurélien Macé, Sergio Carmona, Stefano Ongarello, Jilian A. Sacks, Claudia M. Denkinger.

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
