## [Editor Report · Decision Letter 0]

4 Mar 2021

Dear Dr Denkinger, 

Thank you for submitting your manuscript entitled "The accuracy of novel antigen rapid diagnostics for SARS-CoV-2: a living systematic review and meta-analysis." for consideration by PLOS Medicine.

Your manuscript has now been evaluated by the PLOS Medicine editorial staff as well as by an academic editor with relevant expertise and I am writing to let you know that we would like to send your submission out for external assessment.

Once your full submission is complete, your paper will undergo a series of checks in preparation for external assessment. 

Kind regards,

Richard Turner, PhD

rturner@plos.org

---

## [Decision Letter · Decision Letter 1]

12 May 2021

Dear Dr. Denkinger,

Thank you very much for submitting your manuscript "The accuracy of novel antigen rapid diagnostics for SARS-CoV-2: a living systematic review and meta-analysis." (PMEDICINE-D-21-01004R1) for consideration at PLOS Medicine. 

Your paper was discussed with an academic editor with relevant expertise and sent to independent reviewers, including a statistical reviewer. The reviews are appended at the bottom of this email and any accompanying reviewer attachments can be seen via the link below:

[LINK]

In light of these reviews, we will not be able to accept the manuscript for publication in the journal in its current form, but we would like to invite you to submit a revised version that addresses the reviewers' and editors' comments fully. You will appreciate that we cannot make a decision about publication until we have seen the revised manuscript and your response, and we expect to seek re-review by one or more of the reviewers. 

We hope to receive your revised manuscript by Jun 02 2021 11:59PM. Please email us (plosmedicine@plos.org) if you have any questions or concerns.

Please let me know if you have any questions, and we look forward to receiving your revised manuscript shortly. 

Sincerely,

Richard Turner, PhD

rturner@plos.org

Please update your search to the end of March 2021, say.

Please remove the "Summary" on the title page (this could be repurposed in the author summary, below). 

Please combine the "Methods" and "Findings" subsections of your abstract. 

Please add a new final sentence to the combined subsection, which should begin "Study limitations include ..." or similar and should quote 2-3 of the study's main limitations. 

At line 61, please adapt the style of the "Conclusions" subsection of your abstract as follows: "In this study, we found that ... detected most cases ...", or similar. 

After the abstract, we will need to ask you to add a new and accessible "Author Summary" section in non-identical prose. You may find it helpful to consult one or two recent research papers in PLOS Medicine to get a sense of the preferred style.

Noting the "living" element quoted in your title, we would suggest moving the references to the website, currently at line 105 and in the Methods, to a single mention in the Discussion section. Here we suggest noting how frequently the data will be updated. 

Please also explain to the editors how you plan to proceed with any future peer-reviewed analyses of these data.

Throughout the text, please adapt the style of reference call-outs as follows: "...[12,13]." (noting the absence of spaces within the square brackets).

Noting references 17 and 31, for example, please ensure that all citations have full access details. 

Noting the preprints cited, can DOI numbers, for example, be added so that these can be accessed easily?

Thank you for including the PRISMA checklist. May we suggest using PRISMA 2020 (https://doi.org/10.1371/journal.pmed.1003583) in future?

Please break the checklist out into a separate attachment, labelled "S1_PRISMA_Checklist" or similar and referred to as such in the main text. 

In the checklist, please refer to individual items by section (e.g., "Methods") and paragraph number, not by line or page numbers as these generally change in the event if publication. 

Comments from the reviewers:

*** Reviewer #1: 

Estimated Authors,

I've read with great interest your paper on the accuracy of novel antigen rapid diagnostics for SARS-CoV-2.

The paper is well written, and addresses both the potential pros and the actuans cons of these new instruments.

Materials are clear, tables and images included in the main text are appropriate.

Discussion is consistent with the results and analyzes available evidence.

I've no further recommendations and suggest the eventual acceptance of this paper.

*** Reviewer #2:

Alex McConnachie, Statistical Review

Brummer et al report a systematic review and meta-analysis of the diagnostic accuracy of rapid antigen tests for SARS-CoV-2. This review considers the use of statistics in the paper.

Overall, this is very impressive. It reads quite well, and seems to cover most of the elements that are required. My comments are relatively minor.

Line 384 makes a statement that the confidence intervals for the pooled sensitivity in symptomatic and asymptomatic patients are overlapping. I do not like this - overlapping Cis does not necessarily mean that there is no evidence of a difference between the two groups. It would be better to report a p-value, or a confidence interval for the difference in sensitivity. The same principle applies to other subgroup analyses - when looking at subgroups, I want to know whether there is a difference in the quantity being estimated (e.g. sensitivity) between the subgroups.

The subheading on line 408 mentions viral load, though the subgroup analysis by viral load is reported in the previous section.

Lines 423-426 mention publication bias, but this is very brief. P-values are given, but no interpretation. In what way are the results biased? Can the pooled estimates be corrected for this bias in some way? How does this affect the overall results?

Finally, the forest plots do not include pooled estimates, or measures of heterogeneity. Would these be worth showing here?

*** Reviewer #3: 

Review of the paper entitled

The accuracy of novel antigen rapid diagnostics for SARS-CoV-2: a living systematic review and meta-analysis.

General comment:

The topic of this article is important as there is a great need to follow the improvement in the quality of lateral-flow rapid antigen tests for COVID-19, as it has been done for malaria rapid tests by FIND and WHO and other infectious diseases rapid tests by WHO (HIV, HBV..). Having a living review is very innovative and welcome.

For such an entreprise, there is a need for a well defined reference method that allows to comparing one brand or test version to another. This method can be based either on samples of patients with a pre-defined distribution (or at least a pre-defined threshold) of viral loads measured by quantitative PCR or on laboratory prepared samples containing defined concentration of antigen. No international standard for the validation of COVID RDT is unfortunately available yet. Therefore, the present review has to target studies that include patients whose viral load distribution is very variable from one study to the other, which limits strongly the comparability between studies and the interpretation of pooled results. Stratifying results by viral load thresholds, as authors have done, is therefore key and the main results should be these stratified results (using also a threshold of <20 would have been ideal) rather than the overall result.

Specific comments:

Abstract: 

In "Results", in addition to the pooled sensitivity for Ct-values <30 (significant viral loads), the pooled sensitivity for Ct<25 (moderate and high viral loads) should be added, as it is an important result as mentioned above under general Comment, mentioned by the way by the authors in "Methods". If available, it would be good to also have the result for Ct<20 (high viral loads).

In "Results", in order to show the performance of RDTs in usual testing conditions, e.g. using the good quality brands (SD Biosensor/Roche or Abbott), on the best sample type (naso-pharyngeal), during the first week of symptom (vast majority of patients seen in testing centers), it would be good to add the pooled sensitivity found in the corresponding relevant studies, if feasible. 

In "Conclusion", replace the word "screening" by "diagnosis" as you speak here about symptomatic patients. Also RDT do detect more than "most cases with high viral load". Indeed, they detect most cases with any viral load and the vast majority of patients with significant viral load (a viral load equivalent to a CT of 29 is not a high viral load, it is still a very low viral load as most of these patients have no cultivable virus and are not contagious).

Introduction: 

Line 92: according to most experts (e.g: Mina MJ, et al. Rethinking Covid-19 Test Sensitivity - A Strategy for Containment. New England Journal of Medicine. 2020;383(22):e120), RDT for SARS-CoV-2 are central to the fight against the epidemic, not just complementary. 

Methods:

Line 143-147: a comment on the lack of international standard on the viral load categories that should be studied, or at least the viral load thresholds, which impede accurate comparisons should be added.

Results:

Line 380-386: Subgroup analysis in symptomatic and asymptomatic patients. Presenting overall results could be misleading as it would suggest that the fact of presenting symptoms or not has a direct influence on the sensitivity, why it could be a confounding factor. Indeed one hypothesis for the observed difference is that the main influencing factor is rather the distribution of viral loads that is different (towards lower loads) in asymptomatic persons than in patients, as well mentioned by authors in the discussion. Therefore it is essential to present results by viral load categories (if available), or at least viral load thresholds, to know if RDT are able to detect significant viral loads (and thus most people who are contagious) also in asymptomatic persons.

Line 395-406: Subgroup analysis by Ct-values. As mentioned in the General comment, these results are key and should even be the primary outcome measures of such a review. I would thus propose to put this chapter as second chapter in the overall Results section.

Discussion:

Line 449: I would also mention here the 2 results with the Ct thresholds of 25 and 30.

Conclusion:

As commented above, a viral load corresponding to a Ct of up to 30 is not a high viral load. Indeed most studies suggest that the Ct threshold to differentiate between a culturable form a non-culturable virus is around 25 and than above that threshold most people do not seem to be contagious anymore.

***

[LINK]

---

## [Decision Letter · Decision Letter 2]

8 Jul 2021

Dear Dr. Denkinger,

Thank you very much for re-submitting your manuscript "The accuracy of novel antigen rapid diagnostics for SARS-CoV-2: a living systematic review and meta-analysis." (PMEDICINE-D-21-01004R2) for consideration at PLOS Medicine.

I have discussed the paper with our academic editor and it was also seen again by one reviewer. I am pleased to tell you that, provided the remaining editorial and production issues are fully dealt with, we expect to be able to accept the paper for publication in the journal.

[LINK]

Please let me know if you have any questions, and we look forward to receiving the revised manuscript.   

Sincerely,

Richard Turner, PhD

rturner@plos.org

Requests from Editors:

We are happy to discuss future publications from this project at a suitable time. 

Please remove "The" from the start of the title, and capitalize "A" immediately following the colon. 

We note that you quote the pooled specificity with 95% CI at line 552, for example, and suggest that you also do so in the abstract.

In the abstract, we suggest removing "... making it difficult to draw conclusions from.".

At line 61 we suggest "... cases of SARS-CoV-2 infection" or similar.

At line 181, please make the data extraction file available or remove the statement, noting PLOS' policy on "data not shown" (https://journals.plos.org/plosmedicine/s/data-availability).

At line 333, we suggest making that "... a competing interest".

At line 337, please make that "fewer than".

Please remove the trademarks at lines 386, 411 and 506, and all other instances (including display items).

Data "exists" at line 614?

At line 618, again we suggest adding "... for detection of SARS-CoV-2 infection" or similar.

Please use the general style "... 5 randomly selected papers" throughout the text, although numbers should be spelt out at the start of sentences. 

In the reference list, please use the journal name abbreviation "PLoS ONE".

Can references 271 & 278 be updated?

Comments from Reviewers:

*** Reviewer #2: 

Alex McConnachie, Statistical Review

I thank the authors for their consideration of my original points.

On the issue of talking about overlapping CIs, I have read the Amrhein paper, and whilst I would not consider this to be "guidance", I think the point of the paper is not that p-values or CIs should not be reported, simply that they should not be interpreted as binary indicators of a difference or not. I still think a p-value is a good way to describe the strength of evidence for a difference between subgroups, and a point estimate and CI is a good way to describe the magnitude of a difference. My point was more about whether it is OK to draw conclusions from whether two CIs are overlapping or not. A p-value, or better, an estimate and CI for the difference between subgroups, is a little more informative than a statement about CIs overlapping. However, this is a minor point, and I do not think the authors are interpreting the data incorrectly.

I take the point about not reporting heterogeneity estimates for meta analyses of diagnostic tests - I had not appreciated the nuances in this setting. However, the paper does report pooled estimates of sensitivity and specificity, so I am less clear as to why these are not added to the forest plots. Nevertheless, I accept that this is not the norm, so I will not insist on it.

I welcome the addition of some commentary on the issue of publication bias, though I note that the authors rely on the "significance" of the Deeks' test p-values as a measure of the magnitude of any bias. There may be the same degree of publication bias for all types of diagnostic test; the fact that the Deeks' tests are not statistically significant within individual subgroups of papers does not mean that there is no publication bias. Looking at the funnel plots, and the Deeks' test regression lines, the extent of publication bias in each subgroup shown is quite plausibly the same. I would remove the comments on lines 520-522 about there being less publication bias for these tests.

***

[LINK]

---

## [Editor Report · Decision Letter 3]

14 Jul 2021

Dear Dr Denkinger, 

On behalf of my colleagues and the Academic Editor, Dr Suthar, I am pleased to inform you that we have agreed to publish your manuscript "Accuracy of novel antigen rapid diagnostics for SARS-CoV-2: a living systematic review and meta-analysis." (PMEDICINE-D-21-01004R3) in PLOS Medicine.

Prior to final acceptance, we suggest spelling out "Ct value" at first use in the abstract.

PRESS

Sincerely, 

Richard Turner, PhD 

rturner@plos.org